# Drug Discovery for Periodontitis Treatment Based on Big Data Mining, Systems Biology, and Deep Learning Methods

Chun-Tse Wang and Bor-Sen Chen *

Laboratory of Automatic Control, Signal Processing and Systems Biology, Department of Electrical Engineering, National Tsing Hua University, Hsinchu 30013, Taiwan; s110061588@m110.nthu.edu.tw
* Correspondence: bschen@ee.nthu.edu.tw

**Abstract:** Periodontitis, a chronic inflammatory oral condition triggered by bacteria, archaea, viruses, and eukaryotic organisms, is a well-known and widespread disease around the world. While there are effective treatments for periodontitis, there are also several shortcomings associated with its management, including limited treatment options, the risk of recurrence, and the high cost of treatment. Our goal is to develop a more efficient, systematic drug design for periodontitis before clinical trials. We work on systems drug discovery and design for periodontitis treatment via systems biology and deep learning methods. We first applied big database mining to build a candidate genome-wide genetic and epigenetic network (GWGEN), which includes a protein-protein interaction network (PPIN) and a gene regulatory network (GRN) for periodontitis and healthy control. Next, based on the unhealthy and healthy microarray data, we applied system identification and system order detection methods to remove false positives in candidate GWGENs to obtain real GWGENs for periodontitis and healthy control, respectively. After the real GWGENs were obtained, we picked out the core GWGENs based on how significant the proteins and genes were via the principal network projection (PNP) method. Finally, referring to the annotation of the Kyoto Encyclopedia of Genes and Genomes (KEGG) pathways, we built up the core signaling pathways of periodontitis and healthy control. Consequently, we investigated the pathogenic mechanism of periodontitis by comparing their core signaling pathways. By checking up on the downstream core signaling pathway and the corresponding cellular dysfunctions of periodontitis, we identified the fos proto-oncogene, AP-1 Transcription Factor Subunit (FOS), TSC Complex Subunit 2 (TSC2), Forkhead Box O1 (FOXO1), and nuclear factor kappa-light chain enhancer of activated B cells (NF-κB) as significant biomarkers on which we could find candidate molecular drugs to target. To achieve our ultimate goal of designing a combination of molecular drugs for periodontitis treatment, a deep neural network (DNN)-based drug-target interaction (DTI) model was employed. The model is trained with the existing drug-target interaction databases for the prediction of candidate molecular drugs for significant biomarkers. Finally, we filter out brucine, disulfiram, verapamil, and PK-11195 as potential molecular drugs to be combined as a multiple-molecular drug to target the significant biomarkers based on drug design specifications, i.e., adequate drug regulation ability, high sensitivity, and low toxicity. In conclusion, we investigated the pathogenic mechanism of periodontitis by leveraging systems biology methods and thoroughly developed a therapeutic option for periodontitis treatment via the prediction of a DNN-based DTI model and drug design specifications.

**Keywords:** periodontitis; drug discovery; systems biology; DNN-based DTI model; drug design specifications; KEGG pathways; DTI databases; big data mining



## 1. Introduction

Periodontitis, commonly known as gum disease, is a chronic inflammatory condition that affects the soft and hard tissues around teeth, leading to progressive destruction of the periodontal supporting structures. It is a very common health problem worldwide. In 2019, there were 1.1 billion cases of severe periodontitis globally. According to the Centers

for Disease Control and Prevention (CDC), approximately 47.2% of adults in the United States have some form of periodontitis [1]. The prevalence of periodontitis is generally higher in older adults, males, smokers, and individuals with certain medical conditions such as diabetes, heart disease, and stroke [2]. It is worth noting that the prevalence of periodontitis may be underestimated, as many individuals may not be aware they have the condition or may not seek dental care to receive a diagnosis [3,4].

Plaque is a sticky film of bacteria and food particles that forms on the teeth [5], and if not removed by regular brushing and flossing, it can harden into tartar, which is more difficult to remove (it can only be removed by a dental professional) [5,6]. Tartar buildup can irritate the gums, causing them to become red, swollen, and prone to bleeding [7]. Plaque and tartar also lead to the formation of pockets between the teeth and gums, allowing harmful bacteria to grow. The bacteria in plaque and tartar produce toxins that can irritate the gums and cause inflammation, leading to gingivitis, the early stage of periodontitis. Gingivitis can be reversed with good oral hygiene habits, such as brushing twice a day, flossing daily, and regular dental check-ups. If left untreated, the inflammation can spread to the deeper tissues and bones that support the teeth, causing the pockets to become deeper and more difficult to clean [8]. Chronic periodontitis is the most common form of periodontal disease and is characterized by the inflammation and destruction of the tissues surrounding the teeth. It can lead to tooth loss if left untreated [9].

The symptoms of periodontitis include bad breath, pain or sensitivity in the teeth, receding gums, pus between the teeth and gums, and loose teeth [10]. A dental professional can diagnose periodontitis through a thorough examination of the teeth, gums, and surrounding tissues. The examination may include measuring the depth of the pockets between the teeth and gums using a special probe. X-rays may also be taken to assess the amount of bone loss.

Recently, the primary goal of periodontitis treatment has been to control the infection, prevent further damage to the gums and teeth, and restore oral health. Non-surgical treatments for periodontitis include scaling and root planning, which involve deep cleaning of the teeth and gums to remove bacteria and tartar [11]. In some cases, surgical treatment may be necessary to treat periodontitis. These may include flap surgery, which involves lifting the gums to remove tartar and bacteria, and bone and tissue grafts to regenerate lost bone and tissue [12]. Antibiotics can be prescribed to reduce the bacteria associated with periodontitis or suppress the destruction of the tooth's attachment to the bone [13,14]. Antibiotics can help eliminate the harmful bacteria that are causing gum disease. Commonly used antibiotics for periodontitis include tetracyclines, metronidazole, and amoxicillin. Anti-inflammatory drugs, such as nonsteroidal anti-inflammatory drugs (NSAIDs), may also be prescribed to help reduce the inflammation and pain associated with periodontitis [15]. NSAIDs work by blocking the production of prostaglandins, which are chemicals that cause inflammation and pain in the body. Examples of NSAIDs include ibuprofen and aspirin. Additionally, dual drug delivery could be used for periodontal treatment. Examples include in situ forming gel (ISFG) loaded with doxycycline hyclate and ibuprofen [16], and in situ forming matrix (ISFM) loaded with vancomycin hydrochloride (VH) and borneol [17]. Another approach involves the use of drug-eluting implants, which are placed directly into the periodontal pocket [18]. Although antibiotics and anti-inflammatory drugs can be effective in improving clinical outcomes and slowing down the progression of the disease (the most commonly used systemic antibiotics include amoxicillin, metronidazole, and doxycycline), indiscriminate use of antibiotics can lead to the development of antibiotic-resistant strains of bacteria as well as other adverse effects [19]. Developing a new drug is a complex and time-consuming process that typically takes 13 to 15 years and involves a significant financial investment (2 to 3 billion US dollars from lab to market). In addition, the success rate of drug development is relatively low, with many potential drugs failing to make it past early-stage testing [20,21]. On the other hand, with the increase in antibiotic resistance among periodontal pathogens, the primary goal of periodontitis therapy has shifted to restoring homeostasis in the oral microbiota and its

harmonious balance with the host periodontal tissue [22]. Therefore, a progressive and efficient systematic drug design for the therapeutic treatment of periodontitis is needed. In recent years, deep learning methods have been more commonly used in the prediction of molecular biological systems. Drug-target interaction (DTI) methods are techniques used to study the interactions between drugs and their target molecules. The DNN-based DTI model mainly includes computer algorithms to predict the interactions between drugs and targets (biomarkers), which is generally a binary classification problem with the utilization of drug and target features as inputs. Based on the DNN-based DTI model, we utilized a drug discovery strategy to select candidate molecular drugs for biomarkers of periodontitis treatment [23].

We applied systems biology methods based on genome-wide microarray data to investigate the pathogenic mechanism of periodontitis [24]. The first step is the construction of the candidate GWGEN via big data mining. Next, we utilized the combination of system identification and system order detection by the corresponding microarray data to delete false-positive interactions and regulations from candidate GWGEN. After deleting the false positives, we obtained the real GWGENs of periodontitis and healthy control [25]. Since the annotation of KEGG pathways is at most 6000 proteins, we extract core GWGENs from real GWGENs by selecting the top 6000 significant proteins via the principal network projection (PNP) method. Then, referring to the annotation of KEGG pathways, we could build up the core signaling pathways of periodontitis and the corresponding healthy control to compare their differences [26]. Based on the comparisons, we investigated the pathogenic mechanism of periodontitis, involving complex immune/inflammatory cascades. We analyzed the relationships between proteins, genes, and microenvironments in the core signaling pathways and their downstream cellular dysfunctions. Finally, we selected the fos proto-oncogene, AP-1 transcription factor subunit (FOS), TSC complex subunit 2 (TSC2), forkhead box O1 (FOXO1), and nuclear factor kappa-light chain enhancer of activated B cells (NF-κB) as significant biomarkers for pathological treatment of periodontitis. We applied the features of significant biomarkers and molecular drugs to the well-trained DNN-based DTI model to predict candidate drugs for each biomarker based on the interaction probability between drugs and their targets (biomarkers) [27]. We then filtered out brucine, disulfiram, verapamil, and PK-11195 as potential molecular drugs according to adequate regulation ability, low toxicity, and high sensitivity. A multiple-molecule drug could be selected from these four potential molecular drugs for the therapeutic treatment of periodontitis. Brucine is a natural compound found in the seeds of the Nux-vomica plant. It has both bioactive and toxic properties and is considered a weak alkaline indole alkaloid. Recent studies in pharmacology and clinical practice have shown that brucine has a variety of potential pharmacological effects, including anti-tumor, anti-inflammatory, analgesic, and effects on the cardiovascular and nervous systems [28]. Disulfiram is a medication used in the treatment of alcoholism. The pharmacological effects of disulfiram are primarily related to its ability to produce an aversive reaction to alcohol. This helps discourage individuals from drinking by making the experience unpleasant [29]. The pharmacological effects of verapamil are primarily related to its ability to reduce blood pressure and alleviate the symptoms of angina. It can also be used to treat certain cardiac arrhythmias [30]. The pharmacological effects of PK-11195 are primarily related to its ability to modulate the immune response in the brain. It has been shown to have anti-inflammatory and neuroprotective effects and may have potential therapeutic applications in conditions such as Alzheimer's disease and Parkinson's disease [31].

## 2. Results

### 2.1. Overview of Systems Biology Method for Pathology Mechanism and Systematic Drug Discovery and Design for Periodontitis Treatment

In our work, we applied the systems biology method in Section 4.4 first to investigate the pathogenic mechanism details of periodontitis through microarray data. Next, we selected significant biomarkers of pathogenic mechanisms as drug targets for periodontitis

and then trained the DNN-based DTI model using DTI databases to predict candidate molecular drugs for these drug targets.

We finally screened the desired drugs for potential therapeutic treatment of periodontitis. The screening process is based on how they restore cellular dysfunctions and meet the drug design specifications. The flowchart of the whole system's drug design process is shown in Figure 1.

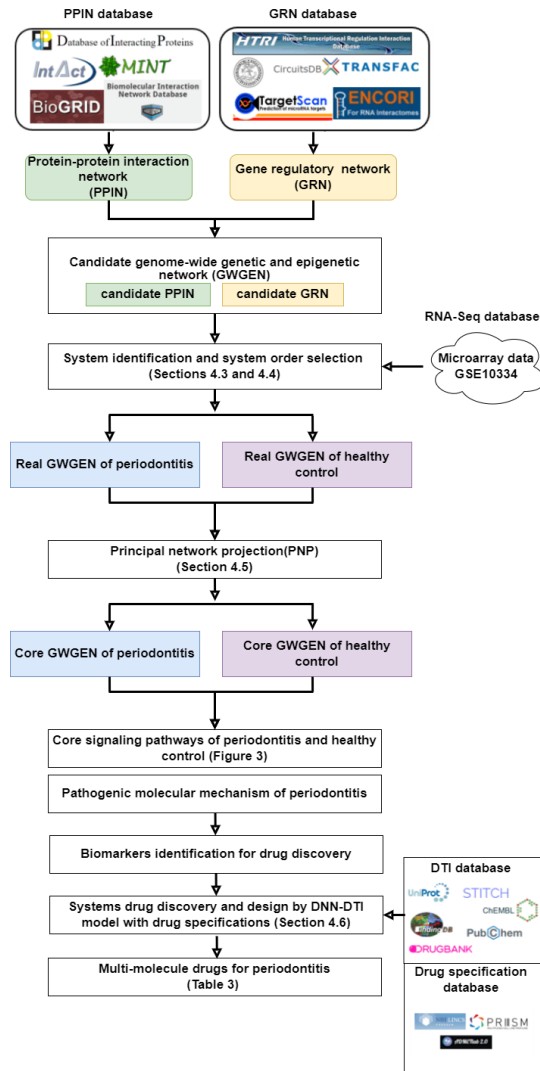

**Figure 1.** The flowchart of the systems biology method and systematic drug design. The construction of candidate GWGEN is obtained by the big data mining method. Followed by the construction of real GWGENs, core GWGENs by principal network projection (PNP) method, and core signaling pathways by the annotation of KEGG pathways for periodontitis and healthy control. We investigate the pathological mechanisms of periodontitis to identify the biomarkers as drug targets. Then DNN-based DTI model can be employed to predict candidate molecular drugs for the biomarkers of periodontitis treatment. Finally, we select a potential multiple molecule drug for periodontitis based on drug design specifications.

The first step is the construction of candidate GWGEN via big data mining from the following databases: BioGRID [32], CircuitDB [33], DIP [34], HTRIdb [35], IntAct [36], ITFP [37], MINT [38], starBase [39], TargetScanHuman [40], and TRANSFAC [41]. Next, we applied the corresponding genome-wide microarray data by system identification and system order detection methods to construct real GWGENs in Figure 2 for periodontitis and healthy control. In Table 1 the total number of nodes in real GWGEN was significantly

reduced compared to candidate GWGEN by pruning off the false positives via the system order detection approach. Even when the scale of both GWGENs was scaled down, the complexity of real GWGENs was still high for pathway annotation by KEGG pathways. In order to fit the KEGG pathway analysis criteria of 6000 nodes at most, we utilized the principal network projection (PNP) method in Section 4.5 to extract the core GWGENs of periodontitis and healthy control from their corresponding real GWGENs. Node extraction was based on their corresponding significant projection values. We selected the top 6000 ranked nodes that construct significant networks in the real GWGEN of periodontitis and healthy control. We classified the significant nodes in core GWGENs into protein, receptor, TF, miRNA, and LncRNA and listed their identified node numbers in Table 1. The top 6000 ranked gene symbol lists of the core signaling pathways of periodontitis and healthy control were organized and projected to the KEGG signaling pathways; we then analyzed and compared the two core signaling pathways. From the enrichment analysis of the database for annotation, visualization, and integrated discovery (DAVID), the critical pathways for core GWGENs of periodontitis and healthy control could be obtained. Based on the KEGG pathway results and referring to related studies that indicated abnormal regulation of downstream genes and cellular dysfunction, we constructed the common and specific core signaling pathways in Figure 3 for periodontitis and healthy control.

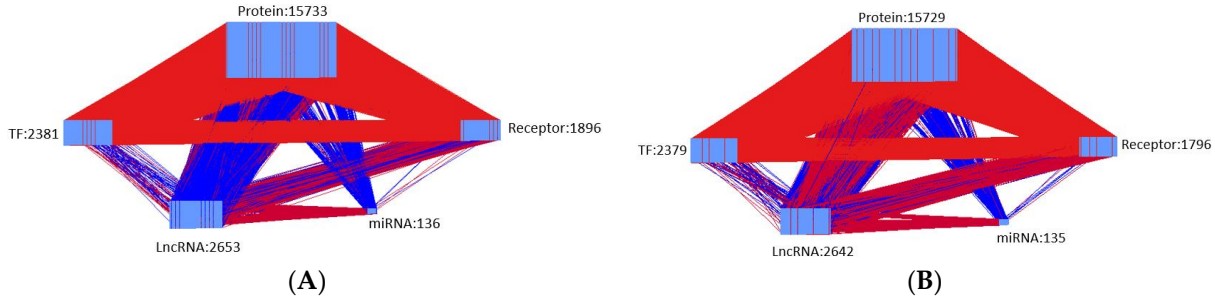

**(A)**                    **(B)**

**Figure 2.** (**A**) The real GWGEN of periodontitis. (**B**) The real GWGEN of healthy control. The labeled numbers indicate the number of nodes. Red lines indicate the interactions between proteins, and blue lines indicate the regulation between genes.

**Table 1.** The first part of the table indicates comparisons of number of nodes in candidate GWGEN, real GWGEN of periodontitis, and real GWGEN of non-periodontitis (healthy control) after system identification. The second part of the table shows the top four pathways of core GWGENs of periodontitis and non-periodontitis based on the enriched DAVID analysis. We utilized these signaling pathways for the construction of core signaling pathways of periodontitis and healthy control in Figure 3.

| Nodes | Candidate GWGEN | Real GWGEN of Periodontitis | Real GWGEN of Healthy Control |
|---|---|---|---|
| Protein | 20,040 | 15,733 | 15,729 |
| Receptor | 2215 | 1896 | 1796 |
| TF | 2395 | 2381 | 2379 |
| miRNA | 153 | 136 | 135 |
| LncRNA | 3315 | 2653 | 2642 |
| Total nodes | 28,118 | 22,799 | 22,681 |
| **Rank** | **Periodontitis KEGG Pathway** | | **Non-Periodontitis KEGG Pathway** |
| 1 | Cell Cycle ($p$-Value = $1.9 \times 10^{-12}$) | | Cell Cycle ($p$-Value = $1.9 \times 10^{-16}$) |
| 2 | FOXO Signaling Pathway ($p$-Value = $2.4 \times 10^{-7}$) | | Insulin Signaling Pathway ($p$-Value = $1.8 \times 10^{-7}$) |
| 3 | Autophagy-Animal ($p$-Value = $2.3 \times 10^{-6}$) | | mRNA Surveillance Pathway ($p$-Value = $7.7 \times 10^{-7}$) |
| 4 | Apoptosis ($p$-Value = $2.9 \times 10^{-6}$) | | FOXO Signaling Pathway ($p$-Value = $2.9 \times 10^{-5}$) |

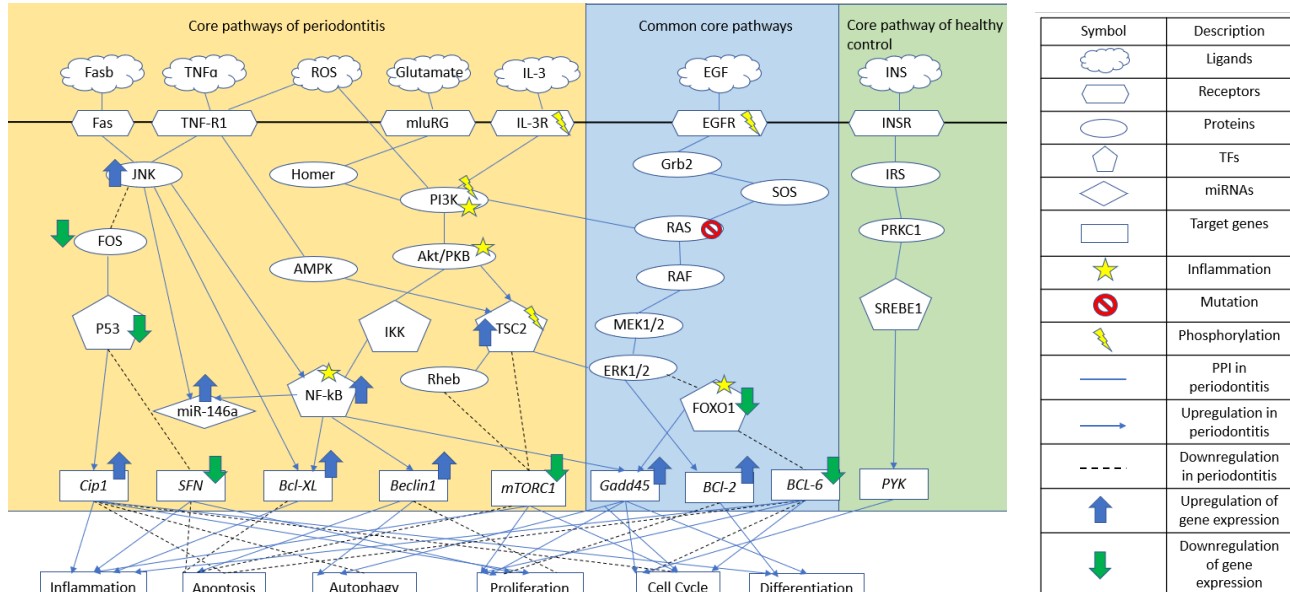

**Figure 3.** The common and specific core signaling pathways between periodontitis and healthy control. The core signaling pathways of periodontitis and healthy control are based on the annotation of core GWGENs derived from KEGG pathways. The light-yellow block contains the specific core signaling pathways of periodontitis. The blue block contains the overlapping, i.e., common core signaling pathways between periodontitis and healthy control. The green block contains the specific core signaling pathway of non-periodontitis healthy control.

## 2.2. Comparing Core Signaling Pathways of Periodontitis and Healthy Control to Identify Biomarkers of Pathological Mechanism of Periodontitis

Untreated inflammation of the gums leads to periodontitis [42]. The following factors can contribute to the development of periodontitis: bacteria in plaque, smoking, poor oral hygiene, genetic predisposition, hormonal changes (such as during pregnancy or menopause), certain medications, and certain medical conditions that affect the immune system (such as diabetes, cardiovascular disease, and respiratory tract infections). We indicate that factors such as TNF-α, Fas-L, glutamate, IL-3, and EGF are produced by the microenvironment of gingival tissue, and the abnormal expression of these factors leads to cellular dysfunction (Figure 3) [43,44]. Starting from the upper left of Figure 3, elevated levels of the ligand TNF-α from multiple cells in gingival tissue are associated with periodontal tissue damage [45]. Membrane TNF receptor type 1 (TNF-R1) mediates the effects of TNF-α. TNF-R1 is expressed in various cells, and its overexpression is involved in the initiation and aggravation of inflammatory responses [46]. Fas-L regulates gingival tissue cell apoptosis, which is highly correlated with periodontitis [47]. The destruction of the host's immune responses and the involvement of reactive oxygen species (ROS) are the main causes of periodontal tissue damage [48]. Experimental evidence shows that ROS formation increases when *Fusobacterium nucleatum* and *Porphyromonas gingivalis* are treated with cells [49]. ROS acts as a double-edged sword since they initiate protective stress responses; however, excessive ROS production decreases antioxidant levels. The imbalance between ROS and the antioxidant defense system leads to oxidative responses that initiate periodontal damage [50].

Ligands ROS and TNF-α bind with TNF-R1, and ligand Fasb binds with Fas; they both activate JNK, initiating cell apoptosis [51]. The activation of JNK causes FOS to downregulate in periodontitis tissue, leading to low expression of the P53 protein, which is one of the reasons for delayed cell death [52]. The upregulation of target gene Cip1 occurred in cells undergoing P53-associated apoptosis [53], and the DNA damage of target gene Cip1 leads to increased levels of Cip1, which is an inhibitor of cell-cycle progression. Low

expression of p53 also leads to downregulation of the target gene SFN, suppressing cell apoptosis and enhancing cell proliferation [54]. Ligand ROS initiates a signaling pathway involved in inflammatory and immune responses by activating NF-κB through signaling proteins P13k, Akt/PKB, and IKK, causing the upregulation of Beclin and Bcl-XL [55], and they are involved in periodontal tissue damage. In addition, activations of JNK and NF-κB lead to the upregulation of the microRNA miR-146a, and several studies have reported higher expression levels of miR-146a in patients with periodontitis than in healthy controls [56]. Increased ROS levels activate AMPK via the receptor TNF-R1, which then induces the phosphorylation and activation of TF TSC2, and thus negatively regulates mTORC1 directly or via Rheb [57]. It has been found that TF TSC2 inhibits mTOR signaling, which is a crucial regulator of proliferation and cellular growth. Without mTOR signaling, inflammatory cells cannot regenerate, causing periodontal tissue destruction [58].

The decreased level of ligand glutamate triggers its receptor, mGluR, which activates the phosphorylation of PI3k and then leads to the MAPK signaling (Ras/Raf/MEK/ERK) pathway downstream of TF FOXO1. The FOXO1 signaling pathway regulates periodontal bacteria-epithelial interactions, immune-inflammatory responses, and wound healing [59]. The downregulation of FOXO1 leads to degraded levels of apoptotic (BCL-6), toll-like (TLR-2), and antioxidant genes, which together help control potential pathogenic bacteria [60]. TF FOXO1 is involved in the activation of adaptive immunity in periodontitis. The higher frequency of Bcl-2 expression indicates delayed apoptosis, leading to increased inflammatory cells in periodontal tissues and resulting in progressive periodontal destruction [61]. Activated PI3k also leads to under-expressed AKT, affecting the TFs IKK, NF-κB, and TSC2, which regulate various target genes. Overexpressed NF-κB leads to significantly higher levels of Bcl-xl and Beclin1, in which Bcl-xl is the most abundant Bcl-x protein and functions to inhibit apoptosis [62], and Beclin1 is a well-established regulator of autophagy [63]. In addition, FOXO1 induces the upregulation of Gadd45. It has been found that Gadd45 is involved in dental epithelial cell proliferation, where it regulates cell growth and gene expression [64].

IL-3 and EGF are inflammatory cytokines that have a strong association with periodontitis [65], and their receptors, IL-3R and EGFR, both activate phosphorylation, which regulates inflammation. The expression of PI3K and AKT increased with the progression of inflammation. EGF can stimulate the secretion of collagenase and gelatinase, and the increased activity of collagenase and gelatinase is associated with severe periodontitis [66]. Research has shown that the serum EGF level in patients with periodontitis is significantly higher than that in healthy controls [67]. Abnormal EGF targets EGFR, affecting the downstream signaling proteins Grb2 and SOS and causing the differentiation of the MAPK signaling pathway [68].

### 2.3. Systematic Drug Discovery Based on Deep Neural Network-Based Drug-Target Interaction Model for Periodontitis Treatment

The investigation of core signaling pathways in Figure 3 lets us gain a thorough understanding of the pathological mechanism of periodontitis. Referring to relevant research, we select the receptors FOS, TFs NF-κB, TSC2, and FOXO1 as significant biomarkers. Our choices are based on whether they are related to cellular dysfunctions of the immune-inflammatory response, apoptosis, cell cycle, or proliferation, and the final goal is to restore their expression levels. Downregulation of FOS leads to abnormal expression of P53, which is correlated with cell destruction. FOS was recently discovered as a treatment option for systemic infections caused by multidrug-resistant bacteria [69]. Overexpression of NF-κB leads to autophagy, which is strongly linked to the inflammatory response [70,71]. Activated FOXO1 plays a critical role in periodontal health, as it participates in the maintenance of homeostasis and adaptation to environmental changes [72]. TSC2 downregulates the expression of mTORC1, which plays a crucial role in periodontal damage. Therefore, we need to find molecular drugs that can upregulate the expression of biomarkers FOS and FOXO1 and downregulate biomarkers NF-κB and TSC2 simultaneously.

In the next step of potential molecular drug discovery and multi-molecule drug design for periodontitis treatment, we train the DNN-based DTI model with drug-target interaction data from drug target databases BIDD, ChEMBL, STITCH, KEGG, PubChem, and UniProt in Section 4.6. For the training dataset, we have 801,291 proved and 100,024 unproved drug-target interactions. The drastic difference between the number of proved and unproved interactions may cause poor performance since the training of the DTI model is dominated by the large class data while the minority class data is ignored. We randomly selected the same amount of drug-target interactions as training data.

The features of drug-target interactions were presented in different units, and the datasets obtained from several databases were too complex for the training of the DNN-based DTI model in Figure 4. Therefore, data preprocessing was required, including standardization, principal component analysis (PCA), and descriptor transformation. We performed feature scaling by standardization and then applied PCA for the dimensionality reduction in order to meet the input dimension of DNN. Out of the 1359 features, 996 were obtained and inputted to train the DTI model for candidate drug prediction for significant biomarkers of periodontitis treatment. We then split the processed data into training data for DNN-based model training and testing data for DNN-based DTI model performance validation. The model parameters were updated by the gradient iterative algorithm in (41)–(44) based on the model error between the true binding level and the predicted binding label of drug-target pairs. The binding probabilities between drugs and each biomarker were then predicted by the DNN-based DTI model in Figure 4.

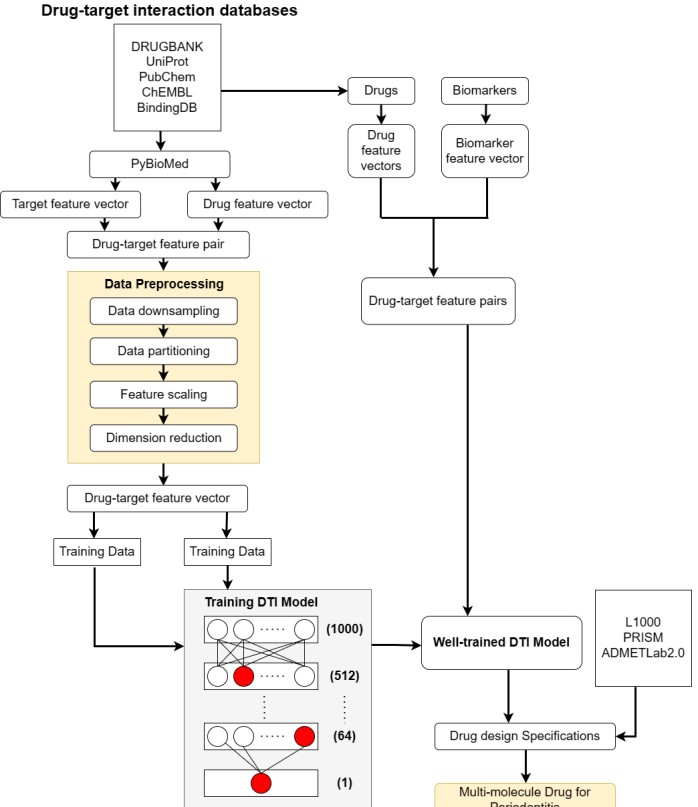

**Figure 4.** The flowchart of the systematic drug design procedure for periodontitis treatment. On the left side of the figure, the drug-target interaction data are obtained from drug-target interaction databases to construct drug-target pair data. After data preprocessing, the data is divided into training data and testing data to train the DNN-based DTI model. On the right side of the figure, the feature vectors of both the biomarkers and the drugs from drug-target interaction databases contain drug-target feature pairs, and they are mounted into the trained DNN-DTI model to predict candidate molecular drugs for the biomarkers. After the prediction, drug design specifications are utilized to filter out the candidate drugs as potential molecular drugs for the treatment of periodontitis.

The architecture of the DNN-based DTI model is shown in Figure 4. It is a fully connected neural network, consisting of an input layer, four hidden layers, and an output layer. For classification issues, each hidden layer is connected by a ReLU function layer, and a sigmoid activation function is added in the output layer. In addition, a dropout layer is added to overcome overfitting. There are 996 nodes in the input layer, with 512, 256, 128, and 64 neurons in the hidden layers, and one node in the output layer. We applied cross-validation to evaluate the performance of the trained DTI model, and the learning curves of loss and accuracy during the model training process were plotted as shown in Figures 5 and 6, respectively. The result of the DNN-based DTI model is a probability value; a higher result value indicates more interaction between the drug and target (biomarker). Additionally, we observed the accuracy of interaction prediction via the receiver operating characteristic (ROC) curve in Figure 7. The value of the area under the curve (AUC) is 0.980, which indicates the model's prediction performance. From Figure 7, we conclude that the prediction performance of the DNN-based DTI model is significantly better than that of the random prediction model (AUC = 0.5). Some candidate drugs for the significant biomarkers FOS, NF-κB, FOXO1, and TSC2 of periodontitis treatment are given in Table 2.

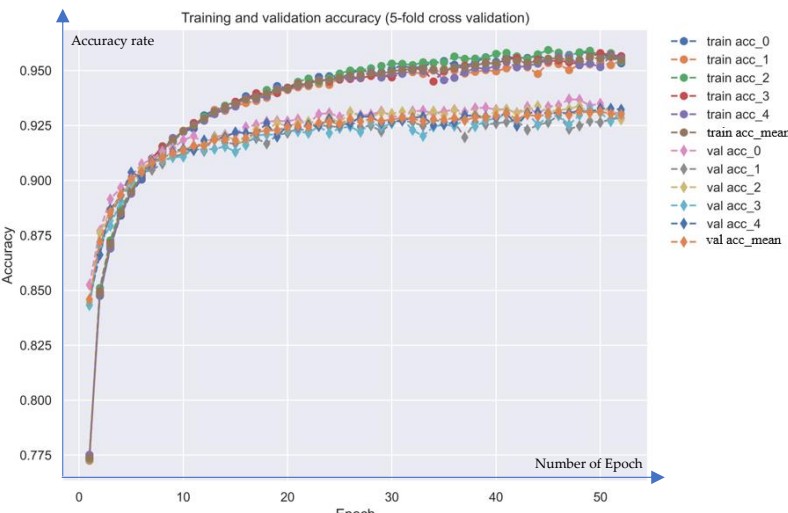

**Figure 5.** Training and validation loss of DNN-based DTI model. (train loss_mean and val loss_mean indicate the model's average loss of training and validation).

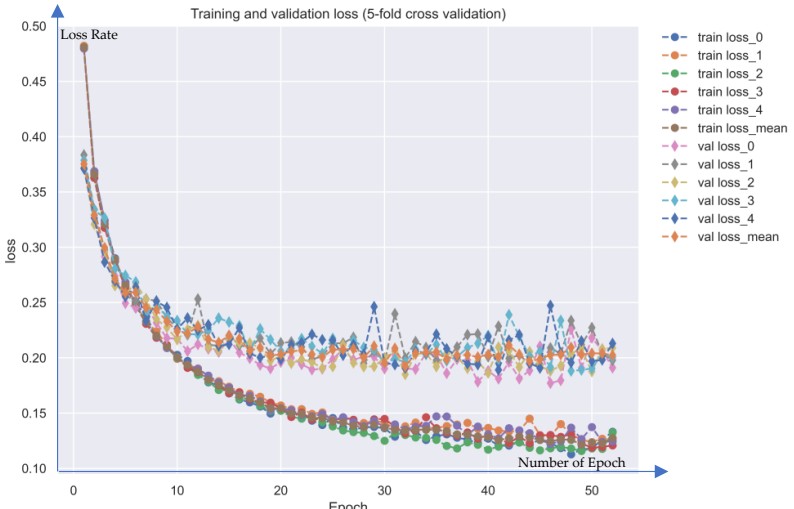

**Figure 6.** Training and validation accuracy of DNN-based DTI model. (train acc_mean and val acc_mean indicate the model's average accuracy of training and validation).

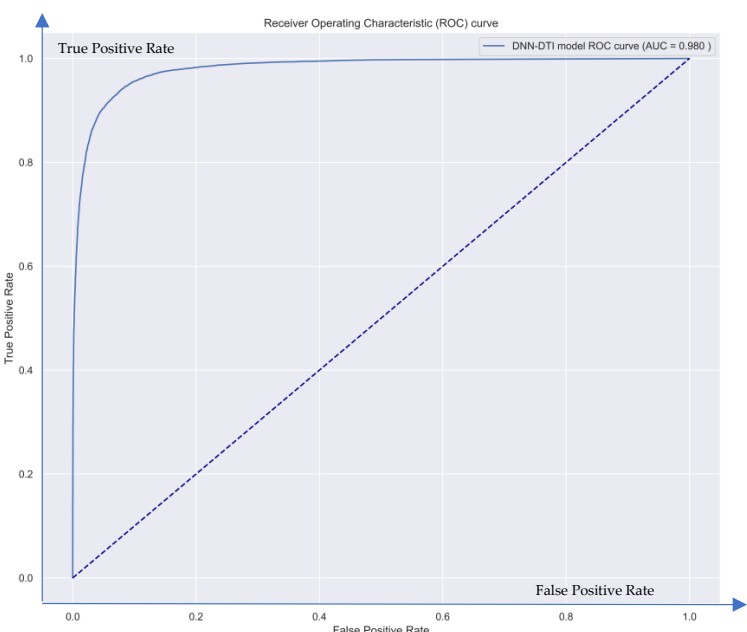

**Figure 7.** The receiver operating characteristic (ROC) curve. The figure shows that the DTI model trained by DNN can achieve an AUC of 0.98. The dotted line indicates the case of AUC = 0.5 as a result of the classifier following random prediction.

**Table 2.** Some candidate molecular drugs for each biomarker of periodontitis treatment from the candidate molecular drugs predicted by DNN-based DTI model and their regulation ability, sensitivity, and toxicity.

| Candidate Drugs | Regulation Ability (L1000) | Sensitivity (PRISM) | Toxicity (LC50, mol/kg) |
|---|---|---|---|
| FOS(−) | | | |
| brucine | 0.337173 | −0.37421 | 5.564 |
| terfenadine | 0.360951 | −0.21913 | 5.437 |
| sulfaphenazole | 0.255523 | −0.32539 | 3.044 |
| alfuzosin | 1.314529 | 0.210014 | 3.945 |
| NF-κB(+) | | | |
| imipramine | −0.35669 | −0.10246 | 4.588 |
| terfenadine | −0.76653 | −0.74069 | 5.437 |
| SB-218078 | −0.72856 | −0.2355 | 4.24 |
| verapamil | 0.048613 | −0.06036 | 6.13 |
| FOXO1(−) | | | |
| disulfiram | 0.073034 | 0.433672 | 8.023 |
| indinavir | 1.023422 | 0.248016 | 4.009 |
| E-4031 | 0.511786 | 0.049582 | 3.265 |
| triclosan | 0.605751 | 0.064844 | 5.892 |
| TSC2(+) | | | |
| erastin | −0.04114 | 0.418918 | 5.621 |
| PK-11195 | −0.28626 | −0.49959 | 5.578 |
| phenothiazine | −0.05596 | −0.43309 | 4.718 |
| loperamide | −0.37009 | 0.003406 | 4.076 |

Finally, we applied drug design specifications to ensure the safety and stability of candidate drugs in Table 2, as predicted by the DNN-based DTI model. To observe the regulatory ability of drug candidates, the library of integrated network-based cellular signatures (LINCS) L1000 Level 5 dataset was downloaded [73]. The L1000 Lenet5 dataset contains 978 genes treated with 19,811 small-molecule compounds in 78 different cell lines. In the resulting ability data, we can identify whether the target gene expression is upregulated (positive) or downregulated (negative) to restore its normal expression. Based on the list, we selected molecular drugs that are capable of restoring the expression value of biomarkers simply by turning them back to their normal expression. For drug sensitivity consideration, we gather drug susceptibility data from the PRISM repurposing dataset [74]. The PRISM dataset includes 4518 drugs tested in 578 human cell lines. The priority is based on how close the sensitivity values are to zero, which indicate cell line sensitivity to chemical perturbations. Another consideration is the lethal concentration (LC50) value, utilizing the ADMETlab 2.0 tool [75]. A higher value of LC50 indicates lesser toxicity to the body of a molecular drug, usually with fewer side effects as well. Finally, based on their reliability, sensitivity, and toxicity, a number of drugs were chosen from candidate molecular drugs for each biomarker of periodontitis treatment (Table 2) based on these drug design specifications. Some candidate molecular drugs for each biomarker of periodontitis treatment were listed in Table 3. We then selected the potential drug combination of brucine, disulfiram, verapamil, and PK-11195 as potential molecule drugs for the therapeutic treatment of periodontitis based on their pharmacological properties.

**Table 3.** The potential multiple-molecule drugs for periodontitis treatment from the candidate molecular drugs in Table 2, based on drug design specifications, i.e., adequate drug regulation ability, high sensitivity and low toxicity. ● denotes the targeting on the corresponding biomarker.

| Potential. Multiple Molecule Drugs | FOS(−) | Regulation Abilities | | TSC2(+) | Sensitivity (PRISM) | Toxicity (LC50, mol/kg) |
|---|---|---|---|---|---|---|
| | | NF-κB(+) | FOXO1(−) | | | |
| brucine | 0.337173● | −0.11899● | 0.423386● | −0.28488● | −0.2282 | 5.564 |
| disulfiram | 0.250077● | −0.26205● | 0.073034● | −0.40304● | −0.34918 | 8.023 |
| verapamil | 0.211182● | 0.048613 | 0.086545● | −0.08621● | −0.06036 | 6.13 |
| PK-11195 | 0.482221● | −0.29659● | 0.205624● | −0.28626● | −0.19207 | 5.578 |

brucine                                           disulfiram

Verapamil                                           PK-11195

## 3. Discussion

The shortcomings of current periodontitis treatment include invasiveness [76], limited efficacy in treating the underlying bacterial infection [77], lack of patient compliance [78], high cost [79], and limited access to specialists. Addressing these issues requires continued research into new and innovative treatments, as well as education and outreach efforts to improve patient compliance and access to care.

Our method strongly differentiates itself from other drug repurposing methods. For example, the paper titled *"Cross-Domain Text Mining to Predict Adverse Events from Tyrosine Kinase Inhibitors for Chronic Myeloid Leukemia"* in [80] explores the use of text mining to predict adverse events associated with tyrosine kinase inhibitors (TKIs) used to treat chronic myeloid leukemia (CML). The authors propose a cross-domain approach, utilizing data from multiple sources, including electronic health records, social media, and scientific literature, to improve the accuracy of adverse event prediction. Our method focuses on building the pathogenic network of periodontitis based on big data mining of the PPIN and GRN databases and microarray data of periodontitis and healthy controls via reverse engineering techniques using a systems biology approach. We compose the core signaling pathways of periodontitis and healthy control to develop a comprehensive and quantitative understanding of the pathogenic mechanism by studying their components, interactions, and emergent properties. This approach involves the use of computational models and system identification (the reverse engineering method), data-driven approaches, and experimental techniques to analyze the pathogenic mechanism of periodontitis. Furthermore, the powerful DNN-based DTI model is trained by DTI databases to predict potential molecular drugs for the significant biomarkers. Overall, systems biology aims to provide a holistic and quantitative understanding of the pathogenic mechanisms of periodontitis, with the ultimate goal of improving the restoration of downstream cellular dysfunctions in periodontitis.

Research has shown that brucine has good anti-tumor effects on breast cancer and hepatocellular carcinoma [81–83]. Brucine N-oxide, a derivative of brucine, has also demonstrated analgesic and anti-inflammatory properties in animal studies. It has been shown to inhibit the activity of certain enzymes involved in the production of inflammatory mediators as well as reduce pain sensitivity in animals [84]. We found out that brucine is able to upregulate the expression of FOS, FOXO1, and NF-κB and, at the same time, downregulate TSC2. Disulfiram is an FDA-approved drug used for the treatment of alcoholism, and it has been found to have antibacterial properties against various strains of bacteria [85]. Studies have shown that disulfiram is a possible treatment for parasitic infections, anxiety disorders, cancer, and latent HIV infection [86]. Disulfiram is proven to be effective in killing various kinds of microbes and can be used as a potential antibiotic therapy [87]. FOS and FOSB were investigated and proven to be risk factors for periodontitis [88]. Moreover, FOS is a promising option to treat systematic infections. Our study discovered the ability of disulfiram to restore the expression of FOS, as well as other biomarkers, to normal expression levels. Verapamil is commonly used to treat high blood pressure, angina (chest pain), and certain heart rhythm disorders [89]. Recent studies based on meta-analysis revealed a significant positive association between periodontitis and increased systolic and diastolic blood pressure and concluded that periodontitis should be considered a potential risk factor for high blood pressure [90,91]. Verapamil strongly measures up to our standards, as it has low sensitivity and a high LC50 value. PK-11195 is a synthetic compound utilized in molecular imaging studies to visualize the peripheral benzodiazepine receptor (PBR) in the body. PBR is found in immune cells, glial cells in the brain, and steroidogenic cells in the adrenal glands and gonads. PK-11195 is strongly linked to inflammation and apoptosis, which are important factors in periodontitis mechanisms [92]. It has been found that PK-11195 has the potential to enhance the effectiveness of chemotherapy in leukemia and myeloma cells [93]. Background studies have shown that oral manifestations, such as gingival bleeding, gingival inflammation or overgrowth, and periodontitis, are the first symptoms of leukemia [94,95].

In conclusion, we selected FOS, TSC2, FOXO1, and NF-κB as significant biomarkers of the pathological mechanism of periodontitis based on systems biology methods. Next, we input the features of biomarkers into the well-trained DNN-based DTI model to predict their candidate molecular drugs. With the systematic drug design method, we selected brucine, disulfiram, verapamil, and PK-11195 from candidate molecular drugs by drug design specifications as drug combinations for potential periodontitis treatment.

## 4. Material and Methods

### 4.1. Systems Biology Methods and Systematic Drug Design for Periodontitis Treatment: An Overview

The procedure of our approach is divided into five steps, as listed below. The complete flowchart is shown in Figure 1.

1.  We obtained data from the genome-wide microarray GSE10334 dataset. For the dataset, ninety subjects with moderate to severe periodontitis (63 with chronic and 27 with aggressive periodontitis) were recruited among those referred to the Columbia University College of Dental Medicine between November 2004 and April 2007. The data is divided into a periodontitis-diseased group and a healthy control group. Next, we constructed candidate GWGEN, including candidate protein-protein interaction network (PPIN) and candidate gene/miRNA/lncRNA regulatory network (GRN), via big database mining.
2.  We identified the real GWGENs for both diseased and healthy control via the system identification method plus the system order detection method in Section 4.4.
3.  We applied the principal network projection method (PNP) to extract the core GWGEN properties, including proteins, receptors, miRNAs, TFs, and lncRNAs, from both real GWGENs to construct core GWGENs in Section 4.5.
4.  Based on the annotation of KEGG pathways, we built up the core signaling pathways of periodontitis as well as the common core pathways of disease and healthy control. Furthermore, we selected biomarkers that play critical roles in pathological mechanisms and lead to downstream cellular dysfunctions in periodontitis.
5.  We built a deep neural network (DNN)-based drug target identification (DTI) model for drug target identification. The DNN-based DTI model is trained by the drug-target interaction database, in which the structures of the drugs and targets are converted into feature vectors. The trained DNN-based DTI model is employed to predict the interaction probability between drugs and their targets (biomarkers), i.e., predict the candidate molecular drugs for biomarkers. We then selected potential molecular drugs for each biomarker from their candidate molecular drugs to combine some potential molecular drugs as a multiple-molecule drug for therapeutic treatment of periodontitis according to drug design specifications.

### 4.2. Data Preprocessing and Big Data Mining for the Construction of Candidate GWGEN

We downloaded genome-wide microarray data from the national center for biotechnology information (NCBI). The genome-wide microarray data include diseased and healthy gingival tissues from 90 non-smoking patients (63 with chronic and 27 with aggressive periodontitis). The patients had no history of systematic periodontal therapy other than occasional prophylaxis and had received no systemic antibiotics or anti-inflammatory drugs for more than 6 months. Furthermore, the patients did not have diabetes or any systemic condition that entails a diagnosis of "Periodontitis as a manifestation of systemic diseases". From the investigation, the downloaded genome-wide microarray dataset GSE10334 contains different numbers of proteins, receptors, miRNAs, TFs, and lncRNAs. We divided the data into 100 diseased samples and 100 healthy controls. Candidate PPIN and GRN, known as the composition of candidate GWGEN, were obtained by big data mining. The candidate PPIN was based on the biomolecular interaction network database (BIND), the biological general repository for interaction datasets database (BioGRID), the database of interacting proteins (DIP), the IntAct molecular interaction database, and the molecular interaction database (MINT). The candidate GRN was based on the CircuitsDB database, the human transcriptional regulation interactions database (HTRIdb), the integrated transcription factor platform database (ITFP), the StarBase2.0 database, the TargetScanHuman database, and the transcription factor database (TRANSFAC). The candidate GWGEN is a Boolean matrix; that is, if there is an interaction between two nodes, we set the value to one, and if there is no interaction, we set it to zero.

*4.3. Construction of the Stochastic System Model to Obtain Real GWGEN of Periodontitis by System Identification Method*

In order to determine real GWGEN for further periodontitis pathway observation, we constructed stochastic interaction and regulatory models for candidate GWGEN. For the protein-protein interaction model of the candidate PPI network, the i-th protein of the n-th sample is described in the following equation:

$$p_i(n) = \sum_{\substack{j=1 \\ j \neq i}}^{E_i} I_{ij} p_i(n) p_j(n) + \zeta_{i,PPIN} + \phi_{i,PPIN}(n)$$

$$\text{for i} = 1, \ldots \text{I, n} = 1, \ldots \text{N} \tag{1}$$

where $p_i(n)$ and $p_j(n)$ indicate the expression level of the i-th protein and the j-th protein in the n-th sample, in which $I_{ij}$ stands for the interaction ability between the i-th and j-th proteins. $E_i$ represents the total protein number interacting with the i-th protein. I stands for the total protein number in the candidate PPIN, while N is the total data sample number. $\zeta_{i,PPIN}$ represents the basal level of the i-th protein expression, owing to the unknown interactions of histone modifications such as phosphorylation and ubiquitination. $\phi_{i,PPIN}(n)$ represents the stochastic noise of the i-th protein in the n-th sample, also denoted as the environment and measurement noise.

Next, for the gene regulatory model, the transcriptional regulation of the q-th gene in the n-th sample can be described in the following gene regulatory equation:

$$g_q(n) = \sum_{f=1}^{R_q} \alpha_{qf} q_f(n) + \sum_{g=1}^{S_q} \beta_{qg} h_g(n) - \sum_{p=1}^{V_q} \delta_{qk} m_k(n) g_q(n) + \chi_q + \tau_q(n)$$

$$\text{for q} = 1, \ldots \text{Q, n} = 1, \ldots \text{N} \tag{2}$$

where $g_q(n)$ denotes the gene expression level for the q-th gene in the n-th sample. The $\alpha_{qf}$ symbol stands for the transcriptional regulatory ability of the f-th TF on the q-th gene. $\beta_{qg}$ symbol represents the transcriptional regulatory ability of the g-th lncRNA on the q-th gene. As for $q_f(n)$, $h_g(n)$, and $m_k(n)$, they separately indicate the gene expression level of the f-th TF, g-th lncRNA, and k-th miRNA in the n-th sample. $\delta_{qk} \geq 0$ shows the post-transcriptional regulatory ability of the k-th miRNA to degrade the miRNA of the q-th gene. Q stands for the total number of genes in candidate GWGEN, and N is the total sample number. $\chi_q$ represents the basal level of q-th gene expression caused by some unknown regulation. If $\chi_q$ is a large value, we need to check the epigenetic regulation. $\tau_q(n)$ represents the gene expression stochastic noise in the q-th gene for the n-th sample.

The expression of the r-th miRNA is affected by the TFs, lncRNAs, and miRNAs, and the transcriptional regulation of the r-th miRNA in the n-th sample can be described in the following gene regulatory equation:

$$k_r(n) = \sum_{f=1}^{R_r} \gamma_{rf} q_f(n) + \sum_{g=1}^{S_r} \varepsilon_{rg} h_g(n) - \sum_{k=1}^{V_r} \mu_{rk} m_k(n) k_r(n) + \chi_r + \tau_r(n)$$

$$\text{for r} = 1, \ldots \text{R, n} = 1, \ldots \text{N} \tag{3}$$

where $k_r(n)$ denotes expression of the r-th miRNA. As for $R_r$ and $S_r$, they separately indicate the total number of TFs and lncRNAs binding to the r-th miRNA, while $V_r$ indicates the total number of miRNAs inhibiting the r-th miRNA. The $\gamma_{rf}$ symbol stands for the transcriptional regulatory ability of the f-th TF on the r-th miRNA. $\varepsilon_{rg}$ symbol represents the transcriptional regulatory ability of the g-th lncRNA on the r-th miRNA. $\geq 0$ shows the post-transcriptional regulatory ability of the k-th miRNA to inhibit the r-th miRNA. $q_f(n)$ indicates the expression of the f-th TFs, $h_g(n)$ indicates the expression of the g-th lncRNA, and $m_k(n)$ indicates the k-th miRNA, respectively. R denotes the total number of miRNAs. $\chi_r$ represents the basal level of the r-th miRNA expression caused by some unknown

regulation. $\tau_r(n)$ represents the gene expression stochastic noise in the r-th miRNA for the n-th sample.

The expression of the u-th lncRNA is affected by the TFs, lncRNAs, and miRNAs, and the transcriptional regulation of the u-th lncRNA in the n-th sample can be described in the following gene regulatory Equation:

$$z_u(n) = \sum_{f=1}^{R_u} \omega_{uf} q_f(n) + \sum_{g=1}^{S_u} \epsilon_{ug} h_g(n) - \sum_{k=1}^{V_u} \vartheta_{uk} m_k(n) z_u(n) + \chi_u + \tau_u(n) \tag{4}$$
$$\text{for u} = 1, \ldots U, \text{ n} = 1, \ldots N$$

where $z_u(n)$ denotes the expression of the u-th lncRNA. As for $R_u$ and $S_u$, they separately indicate the total number of TFs and lncRNAs binding to the u-th lncRNA, while $V_u$ indicates the total number of miRNAs inhibiting the u-th lncRNA. The $\omega_{uf}$ symbol stands for the transcriptional regulatory ability of the f-th TF on the u-th lncRNA. $\epsilon_{ug}$ symbol represents the transcriptional regulatory ability of the g-th lncRNA on the u-th lncRNA. $\delta_{qk} \geq 0$ shows the post-transcriptional regulatory ability of the k-th miRNA to inhibit the u-th lncRNA. $q_f(n)$ indicates the expression of the f-th TF, $h_g(n)$ indicates the g-th lncRNA, and $m_k(n)$ indicates the k-th miRNA, respectively. U denotes the total number of lncRNAs. $\chi_u$ represents the basal level of the u-th lncRNA expression caused by some unknown regulation. $\tau_u(n)$ represents the stochastic noise in the u-th lncRNA for the n-th sample.

### 4.4. Constructing Real GWGENs of Periodontitis and Healthy Control by System Identification and System Order Detection Methods

We constructed PPI models in Equation (1) as well as candidate GRN models for genes in (2), miRNAs in (3), and lncRNAs in (4). The next step is to apply system identification and system order detection methods using real microarray data from periodontitis and healthy controls to prune the false-positive interactions and obtain real GWGENs. The protein interactive model in (1) and gene regulatory model in (2)–(4) can be rearranged as the following linear regression forms, respectively.

$$p_i(n) = [p_i(n)p_1(n) \cdots p_i(n)p_{Ei}(n)1] \times \begin{bmatrix} I_{i1} \\ \vdots \\ I_{iEi} \\ \zeta_i \end{bmatrix} + \phi_i(n) \tag{5}$$

$$= j_i(n) \cdot \theta_i + \phi_i(n) \quad for\ i = 1, \ldots, I,\ n = 1, \ldots, N$$

$$g_q(n) = [q_1(n) \cdots q_{Rq}(n)h_1(n) \cdots h_{Sq}(n)m_1(n)g_q(n) \cdots m_{Vq}(n)g_q(n)1] \times \begin{bmatrix} \alpha_{q1} \\ \vdots \\ \alpha_{qRq} \\ \beta_{qg} \\ \vdots \\ \beta_{qSq} \\ -\delta_{q1} \\ \vdots \\ -\delta_{qVq} \\ \chi_q \end{bmatrix} + \tau_q(n) \tag{6}$$

$$= j_q(n) \cdot \theta_q + \tau_q(n),\ for\ q = 1, \ldots, Q,\ n = 1, \ldots, N$$

$$k_r(n) = [r_1(n)\cdots r_{Rr}(n)h_1(n)\cdots h_{Sr}(n)m_1(n)g_r(n)\cdots m_{Vr}(n)g_r(n)1] \times \begin{bmatrix} \alpha_{r1} \\ \vdots \\ \alpha_{rRr} \\ \beta_{rg} \\ \vdots \\ \beta_{rSr} \\ -\delta_{r1} \\ \vdots \\ -\delta_{rVr} \\ \chi_r \end{bmatrix} + \tau_r(n) \qquad (7)$$

$$= j_r(n)\cdot\theta_r + \tau_r(n), \; for \; r = 1, \ldots, R, \; n = 1,\ldots, N$$

$$z_u(n) = [u_1(n)\cdots q_{Ru}(n)h_1(n)\cdots h_{Su}(n)m_1(n)g_u(n)\cdots m_{Vu}(n)g_u(n)1] \times \begin{bmatrix} \alpha_{u1} \\ \vdots \\ \alpha_{uRu} \\ \beta_{ug} \\ \vdots \\ \beta_{uSu} \\ -\delta_{u1} \\ \vdots \\ -\delta_{uVu} \\ \chi_u \end{bmatrix} + \tau_u(n) \qquad (8)$$

$$= j_u(n)\cdot\theta_u + \tau_u(n), \; for \; u = 1, \ldots, U, \; n = 1,\ldots, N$$

where $j_i(n)$, $j_q(n)$, $j_r(n)$ and $j_u(n)$ are the corresponding regression vectors, $\theta_i$, $\theta_q$, $\theta_r$, $\theta_u$ are the corresponding parameter factors to denote the interactive abilities as protein-protein interaction (PPI) of proteins and the regulatory abilities of genes, miRNAs, and lncRNAs in the genetic regulatory network (GRN). $\phi_i(n)$, $\tau_q(n)$, $\tau_r(n)$ and $\tau_u(n)$ separately denote the environmental noise of the i-th protein, q-th gene, r-th miRNA, and u-th lncRNA in vector forms.

The above linear regression forms (5) to (8) can be denoted, respectively, as below:

$$\begin{bmatrix} p_i(1) \\ p_i(2) \\ \vdots \\ p_i(N) \end{bmatrix} = \begin{bmatrix} j_i(1) \\ j_i(2) \\ \vdots \\ j_i(N) \end{bmatrix} \cdot \theta_i + \begin{bmatrix} \phi_i(1) \\ \phi_i(2) \\ \vdots \\ \phi_i(N) \end{bmatrix} \qquad (9)$$

For $i = 1, \ldots, I, \; n = 1,\ldots, N.$

$$\begin{bmatrix} g_q(1) \\ g_q(2) \\ \vdots \\ g_q(N) \end{bmatrix} = \begin{bmatrix} j_q(1) \\ j_q(2) \\ \vdots \\ j_q(N) \end{bmatrix} \cdot \theta_q + \begin{bmatrix} \tau_q(1) \\ \tau_q(2) \\ \vdots \\ \tau_q(N) \end{bmatrix} \qquad (10)$$

For $q = 1, \ldots, Q, \; n = 1,\ldots, N.$

$$\begin{bmatrix} k_r(1) \\ k_r(2) \\ \vdots \\ k_r(N) \end{bmatrix} = \begin{bmatrix} j_r(1) \\ j_r(2) \\ \vdots \\ j_r(N) \end{bmatrix} \cdot \theta_r + \begin{bmatrix} \tau_r(1) \\ \tau_r(2) \\ \vdots \\ \tau_r(N) \end{bmatrix} \tag{11}$$

For $r = 1, \ldots, R$, $n = 1, \ldots, N$.

$$\begin{bmatrix} z_u(1) \\ z_u(2) \\ \vdots \\ z_u(N) \end{bmatrix} = \begin{bmatrix} j_u(1) \\ j_u(2) \\ \vdots \\ j_u(N) \end{bmatrix} \cdot \theta_u + \begin{bmatrix} \tau_u(1) \\ \tau_u(2) \\ \vdots \\ \tau_u(N) \end{bmatrix} \tag{12}$$

For $u = 1, \ldots, U$, $n = 1, \ldots, N$.

Equations (9)–(12) can be expressed, respectively, in the following algebraic Equations:

$$P_i = \mu_i \cdot \theta_i + \varepsilon_i, \ for \ i = 1, \ldots, I \tag{13}$$

$$G_q = \mu_q \cdot \theta_q + \varepsilon_q, \ for \ q = 1, \ldots, Q \tag{14}$$

$$K_r = \mu_r \cdot \theta_r + \varepsilon_r, \ for \ r = 1, \ldots, R \tag{15}$$

$$Z_u = \mu_u \cdot \theta_u + \varepsilon_u, \ for \ u = 1, \ldots, U \tag{16}$$

where $\mu_i$ represents the regression matrix of proteins, $\mu_q$ represents the regression matrix of genes, $\mu_r$ represents the regression matrix of miRNAs, and $\mu_u$ represents the regression matrix of lncRNA.

Next, we focus on identifying the interactive and regulatory parameters of candidate GWGEN. We utilized the MATLAB optimization toolbox to solve the constrained linear least squares parameter problem to estimate the parameter vectors $\theta_i$ in PPIN, $\theta_q$, $\theta_r$, $\theta_u$ in GRN in candidate GWGEN. The parameter estimations of candidate GWGENs for real GWGENs of periodontitis and healthy control are based on their microarray data as follows:

$$\widehat{\theta_i} = \underset{\theta_i}{\mathrm{argmin}} \frac{1}{2} \|\mu_i \cdot \theta_i - P_i\|_2^2$$

$$\widehat{\theta_q} = \underset{\theta_q}{\mathrm{argmin}} \frac{1}{2} \|\mu_q \cdot \theta_q - G_q\|_2^2 \tag{17}$$

$$\text{subject to} \begin{bmatrix} 0 & \cdots & \cdots & 0 & 0 & \cdots & \cdots & 0 & 1 & 0 & \cdots & 0 & 0 \\ \vdots & \ddots & \ddots & \vdots & \vdots & \ddots & \ddots & \vdots & 0 & \ddots & \ddots & \vdots & \vdots \\ \vdots & \ddots & \ddots & \vdots & \vdots & \ddots & \ddots & \vdots & \vdots & \ddots & \ddots & 0 & \vdots \\ 0 & \cdots & \cdots & 0 & 0 & \cdots & \cdots & 0 & 0 & \cdots & 0 & 1 & 0 \end{bmatrix} \theta_q \le \begin{bmatrix} 0 \\ \vdots \\ \vdots \\ 0 \end{bmatrix} \tag{18}$$

$$\widehat{\theta_r} = \underset{\theta_r}{\mathrm{argmin}} \frac{1}{2} \|\mu_r \cdot \theta_r - K_r\|_2^2$$

$$\text{subject to} \begin{bmatrix} 0 & \cdots & \cdots & 0 & 0 & \cdots & \cdots & 0 & 1 & 0 & \cdots & 0 & 0 \\ \vdots & \ddots & \ddots & \vdots & \vdots & \ddots & \ddots & \vdots & 0 & \ddots & \ddots & \vdots & \vdots \\ \vdots & \ddots & \ddots & \vdots & \vdots & \ddots & \ddots & \vdots & \vdots & \ddots & \ddots & 0 & \vdots \\ 0 & \cdots & \cdots & 0 & 0 & \cdots & \cdots & 0 & 0 & \cdots & 0 & 1 & 0 \end{bmatrix} \theta_r \le \begin{bmatrix} 0 \\ \vdots \\ \vdots \\ 0 \end{bmatrix} \tag{19}$$

$$\widehat{\theta_u} = \underset{\theta_u}{\mathrm{argmin}} \frac{1}{2} \|\mu_u \cdot \theta_u - Z_u\|_2^2$$

$$\text{subject to} \begin{bmatrix} 0 & \cdots & \cdots & 0 & 0 & \cdots & \cdots & 0 & 1 & 0 & \cdots & 0 & 0 \\ \vdots & \ddots & \ddots & \vdots & \vdots & \ddots & \ddots & \vdots & 0 & \ddots & \ddots & \vdots & \vdots \\ \vdots & \ddots & \ddots & \vdots & \vdots & \ddots & \ddots & \vdots & \vdots & \ddots & \ddots & 0 & \vdots \\ 0 & \cdots & \cdots & 0 & 0 & \cdots & \cdots & 0 & 0 & \cdots & 0 & 1 & 0 \end{bmatrix} \theta_u \leq \begin{bmatrix} 0 \\ \vdots \\ \vdots \\ 0 \end{bmatrix} \tag{20}$$

In the above parameter estimation problems, we utilized the constrained least squares parameter estimation in (18)–(20) to ensure that the estimated post-transcriptional regulation parameters of miRNAs are less than or equal to zero. That is, to make sure that the regulation of miRNAs on genes is negative. $\widehat{\theta}_i$, $\widehat{\theta}_q$, $\widehat{\theta}_r$ and $\widehat{\theta}_u$ separately stand for the estimated parameters of the i-th protein, q-th gene, r-th miRNA, and u-th lncRNA, respectively. After that, the parameters of candidate GWGENs for periodontitis and healthy control are identified from their corresponding microarray data, respectively.

Since identified candidate GWGENs contain many false positives, we need to prune these false positives by applying Akaike Information Criteria (AIC) to detect the true interaction number of each protein in PPIN and the true regulation number of each gene in GWGEN by their estimation errors in the above system identification process [96]. We can obtain the real GWGEN after pruning false positives out of system order in GWGEN. The AIC equations for the i-th protein and q-th gene are shown below, respectively,

$$\text{AIC}(E_i) = \log\left(\Psi_i^2\right) + \frac{2(E_i + 1)}{N} \tag{21}$$

where

$$\Psi_i = \sqrt{\frac{\left(P_i - \left(\mu_i \cdot \widehat{\theta}_i\right)\right)^T \left(P_i - \left(\mu_i \cdot \widehat{\theta}_i\right)\right)}{N}} \tag{22}$$

$\Psi_i$ is the estimated residual error in (17), and $E_i$ is the system order (number) of the protein-protein interactions for the previous parameter estimation problem (17), and

$$\text{AIC}\left(R_q,\ S_q,\ V_q\right) = \log\left(\Psi_q^2\right) + \frac{2(\partial_q + 1)}{N} \tag{23}$$

where

$$\Psi_q = \sqrt{\frac{\left(G_q - \left(\mu_q \cdot \widehat{\theta}_q\right)\right)^T \left(G_q - \left(\mu_q \cdot \widehat{\theta}_q\right)\right)}{N}} \tag{24}$$

$\Psi_q$ indicates the estimated residual error for the parameter estimation problem (18). $\partial_q = R_q + S_q + V_q$, which indicates the system-order regulations on the q-th gene. Considering the system order detection method of AIC, the real order of system GWGEN requires minimizing the AIC via the system order detection method shown below:

$$E_i^* = \underset{E_i}{\arg\min}\ AIC(E_i),\ for\ i = 1, \ldots, I \tag{25}$$

$$R_q^*,\ S_q^*,\ V_q^* = \underset{R_q,\ S_q,\ V_q}{\arg\min}\ AIC\left(R_q,\ S_q,\ V_q\right),\ for\ q = 1, \ldots, Q \tag{26}$$

where $E_i^*$ stands for the real number of PPIs (protein-protein interactions) for the i-th protein. $R_q^*$, $S_q^*$, $V_q^*$ separately represent the real number of regulations by TFs, lncRNA, and miRNA on the q-th gene, respectively. The pruning process and parameter optimization for detecting false positives are described as follows: In general, increasing parameter number (system order) will result in good model fit such that the log residual error in the first term of AIC decreases and the second term of AIC increases, and vice versa. Therefore, there should be exact parameter numbers in the system in order to achieve the minimum AIC among all possible combinations for each protein and gene. Forward and backward

stepwise algorithms were both adopted to gain the minimum AIC in Equations (21)–(26) with the help of the lsqlin function in the 2021 MATLAB optimization toolbox. Therefore, we trimmed the insignificant parameters as false positives in the candidate GWGEN out of system order detected by AIC to obtain real GWGENs of periodontitis and healthy control. After the AIC methods in (25) and (26), the protein-protein interactions and regulations that are out of real number (system order $E_i^*$, $R_q^*$, $S_q^*$, $V_q^*$) are identified as false positives in the real GWGEN system. The false positives are to be pruned from candidate GWGENs to obtain real GWGENs of periodontitis and healthy control.

*4.5. Extraction of the Core GWGEN from Real GWGEN for Core Signaling Pathways via Principal Network Projection (PNP) Method*

The real GWGENs for periodontitis and healthy control are still relatively complex for the annotation of pathways by the KEGG pathway, despite the previous step of pruning false positives; i.e., we need to transform real GWGENs into signaling pathways in order to get a better view of the pathogenic mechanisms behind periodontitis. We developed a critical extraction step before pathway annotation of real GWGENs, namely, to extract core GWGENs via the PNP method. We extracted the top 6000 ranked nodes to be annotated by the KEGG pathway. We first introduced the network matrix K of real GWGENs as below:

$$K = \begin{bmatrix} k_{protein \leftrightarrow protein} & 0 & 0 \\ k_{TF \rightarrow gene} & k_{lncRNA \rightarrow gene} & k_{miRNA \rightarrow gene} \\ k_{TF \rightarrow lncRNA} & k_{lncRNA \rightarrow lncRNA} & k_{miRNA \rightarrow lncRNA} \\ k_{TF \rightarrow miRNA} & k_{lncRNA \rightarrow miRNA} & k_{miRNA \rightarrow miRNA} \end{bmatrix} \quad (27)$$

where $k_{protein \leftrightarrow protein}$ stands for the sub-matrix of PPIs, the double-headed arrow indicates that the interaction is bidirectional. The rest of the matrix elements represent the transcriptional regulatory ability sub-matrixes on top of each other. The details are shown below:

$$K = \begin{bmatrix}
\widehat{\tau}_{11} & \cdots & \widehat{\tau}_{1w} & \cdots & \widehat{\tau}_{1W} & 0 & \cdots & 0 & \cdots & 0 & 0 & \cdots & 0 & \cdots & 0 \\
\vdots & \ddots & \vdots & \ddots & \vdots & \vdots & \ddots & \vdots & \ddots & \vdots & \vdots & \ddots & \vdots & \ddots & \vdots \\
\widehat{\tau}_{s1} & \cdots & \widehat{\tau}_{sw} & \cdots & \widehat{\tau}_{sW} & 0 & \cdots & 0 & \cdots & 0 & 0 & \cdots & 0 & \cdots & 0 \\
\vdots & \ddots & \vdots & \ddots & \vdots & \vdots & \ddots & \vdots & \ddots & \vdots & \vdots & \ddots & \vdots & \ddots & \vdots \\
\widehat{\tau}_{S1} & \cdots & \widehat{\tau}_{Sw} & \cdots & \widehat{\tau}_{SW} & 0 & \cdots & 0 & \cdots & 0 & 0 & \cdots & 0 & \cdots & 0 \\
\widehat{\alpha}_{11} & \cdots & \widehat{\alpha}_{1f} & \cdots & \widehat{\alpha}_{1F} & \widehat{\beta}_{11} & \cdots & \widehat{\beta}_{1g} & \cdots & \widehat{\beta}_{1G} & -\widehat{\delta}_{11} & \cdots & -\widehat{\delta}_{1h} & \cdots & -\widehat{\delta}_{1H} \\
\vdots & \ddots & \vdots & \ddots & \vdots & \vdots & \ddots & \vdots & \ddots & \vdots & \vdots & \ddots & \vdots & \ddots & \vdots \\
\widehat{\alpha}_{t1} & \cdots & \widehat{\alpha}_{tf} & \cdots & \widehat{\alpha}_{tF} & \widehat{\beta}_{t1} & \cdots & \widehat{\beta}_{tg} & \cdots & \widehat{\beta}_{tG} & -\widehat{\delta}_{t1} & \cdots & -\widehat{\delta}_{th} & \cdots & -\widehat{\delta}_{tH} \\
\vdots & \ddots & \vdots & \ddots & \vdots & \vdots & \ddots & \vdots & \ddots & \vdots & \vdots & \ddots & \vdots & \ddots & \vdots \\
\widehat{\alpha}_{T1} & \cdots & \widehat{\alpha}_{Tf} & \cdots & \widehat{\alpha}_{TF} & \widehat{\beta}_{T1} & \cdots & \widehat{\beta}_{Tg} & \cdots & \widehat{\beta}_{TG} & -\widehat{\delta}_{T1} & \cdots & -\widehat{\delta}_{Th} & \cdots & -\widehat{\delta}_{TH} \\
\widehat{\varepsilon}_{11} & \cdots & \widehat{\varepsilon}_{1f} & \cdots & \widehat{\varepsilon}_{1F} & \widehat{\gamma}_{11} & \cdots & \widehat{\gamma}_{1g} & \cdots & \widehat{\gamma}_{1G} & -\widehat{\sigma}_{11} & \cdots & -\widehat{\sigma}_{1h} & \cdots & -\widehat{\sigma}_{1H} \\
\vdots & \ddots & \vdots & \ddots & \vdots & \vdots & \ddots & \vdots & \ddots & \vdots & \vdots & \ddots & \vdots & \ddots & \vdots \\
\widehat{\varepsilon}_{u1} & \cdots & \widehat{\varepsilon}_{uf} & \cdots & \widehat{\varepsilon}_{uF} & \widehat{\gamma}_{u1} & \cdots & \widehat{\gamma}_{ug} & \cdots & \widehat{\gamma}_{uG} & -\widehat{\sigma}_{u1} & \cdots & -\widehat{\sigma}_{uh} & \cdots & -\widehat{\sigma}_{11} \\
\vdots & \ddots & \vdots & \ddots & \vdots & \vdots & \ddots & \vdots & \ddots & \vdots & \vdots & \ddots & \vdots & \ddots & \vdots \\
\widehat{\varepsilon}_{U1} & \cdots & \widehat{\varepsilon}_{Uf} & \cdots & \widehat{\varepsilon}_{UF} & \widehat{\gamma}_{U1} & \cdots & \widehat{\gamma}_{Ug} & \cdots & \widehat{\gamma}_{UG} & -\widehat{\sigma}_{U1} & \cdots & -\widehat{\sigma}_{Uh} & \cdots & -\widehat{\sigma}_{UH} \\
\widehat{\varphi}_{11} & \cdots & \widehat{\varphi}_{1f} & \cdots & \widehat{\varphi}_{1F} & \widehat{\rho}_{11} & \cdots & \widehat{\rho}_{1g} & \cdots & \widehat{\rho}_{1G} & -\widehat{\mu}_{11} & \cdots & -\widehat{\mu}_{1h} & \cdots & -\widehat{\mu}_{1H} \\
\vdots & \ddots & \vdots & \ddots & \vdots & \vdots & \ddots & \vdots & \ddots & \vdots & \vdots & \ddots & \vdots & \ddots & \vdots \\
\widehat{\varphi}_{v1} & \cdots & \widehat{\varphi}_{vf} & \cdots & \widehat{\varphi}_{vF} & \widehat{\rho}_{v1} & \cdots & \widehat{\rho}_{vg} & \cdots & \widehat{\rho}_{vG} & -\widehat{\mu}_{v1} & \cdots & -\widehat{\mu}_{vh} & \cdots & -\widehat{\mu}_{vH} \\
\vdots & \ddots & \vdots & \ddots & \vdots & \vdots & \ddots & \vdots & \ddots & \vdots & \vdots & \ddots & \vdots & \ddots & \vdots \\
\widehat{\varphi}_{V1} & \cdots & \widehat{\varphi}_{Vf} & \cdots & \widehat{\varphi}_{VF} & \widehat{\rho}_{V1} & \cdots & \widehat{\rho}_{Vg} & \cdots & \widehat{\rho}_{VG} & -\widehat{\mu}_{V1} & \cdots & -\widehat{\mu}_{Vh} & \cdots & -\widehat{\mu}_{VH}
\end{bmatrix} \quad (28)$$

$$\in \mathbb{R}^{(S^*+T^*+U^*+V^*) \times (F^*+G^*+H^*)}$$

where $\widehat{\tau}_{sw}$ indicates the interaction ability between the s-th protein and the w-th protein, $\widehat{\lambda}_{vg}$ indicates the interaction ability between the v-th lncRNA and the g-th miRNA. The zeros stand for no interaction or regulation after AIC. The estimated parameters in (28)

are obtained by the least squares parameter estimation in (17) and the constrained least squares parameter estimation from (18)–(20), followed by AIC from (23)–(26). The PNP method is based on the following singular value decomposition:

$$\mathrm{K} = \mathrm{S}VD^T \tag{29}$$

where $\mathrm{S} \in \mathbb{R}^{(S^*+T^*+U^*+V^*)\times(S^*+T^*+U^*+V^*)}$, $D^T \in \mathbb{R}^{(F^*+G^*+H^*)\times(F^*+G^*+H^*)}$ are unitary singular matrices. $V = \mathrm{diag}(v_1, \ldots, v_i, \ldots, v_{F^*+G^*+H^*}) \in \mathbb{R}^{(S^*+T^*+U^*+V^*)\times(F^*+G^*+H^*)}$ denotes a diagonal matrix that is consisted of $(F^* + G^* + H^*)$ singular values of matrix K in the descending order, i.e., $v_1 \geq v_2 \geq \ldots \geq v_{F^*+G^*+H^*} \geq 0$. An example for diagonal matrix of $v_2$ and $v_2$ is shown as follows:

$$\mathrm{diag}(v_1, v_2) = \begin{bmatrix} v_1 & 0 \\ 0 & v_2 \end{bmatrix} \tag{30}$$

which is extended to diagonal matrix V:

$$V = \begin{bmatrix} v_1 & 0 & \cdots & 0 & \cdots & 0 \\ 0 & v_2 & \cdots & 0 & \cdots & 0 \\ \vdots & \vdots & \ddots & \vdots & \ddots & \vdots \\ 0 & 0 & \cdots & v_i & \cdots & 0 \\ \vdots & \vdots & \ddots & \vdots & \ddots & \vdots \\ 0 & 0 & \cdots & 0 & \cdots & v_{(F^*+G^*+H^*)} \\ 0 & 0 & \cdots & 0 & \cdots & 0 \\ \vdots & \vdots & \ddots & \vdots & \ddots & \vdots \\ 0 & 0 & \cdots & 0 & \cdots & 0 \end{bmatrix} \tag{31}$$

The eigen expression function $E_i$ for singular value-normalization is denoted below:

$$E_i = \frac{v_i^2}{\sum_{i=1}^{(F^*+G^*+H^*)} v_i^2}, \quad \sum_{i-1}^{(F^*+G^*+H^*)} E_i = 1 \tag{32}$$

Based on energy consideration, we selected the top H normalized singular values of network matrix K of real GWGEN with energy greater or equal to the threshold rate (0.85): $\sum_{i=1}^{H} E_i \geq 0.85$. This denotes the principal network with 85% of the real GWGEN's energy. Next, we project the interactions, i.e., edges of each node of real GWGEN (each row in the network matrix K), onto the top H significant singular vectors as follows:

$$R(a, b) = x_{a,:} \cdot y_{:,b}^T$$
$$\text{for } a = 1, \ldots, S^* + T^* + U^* + V^*, b = 1, \ldots, H \tag{33}$$

$R(a, b)$ stands for the projection value of the a-th node on the b-th significant singular vector. $x_{a,:}$ and $y_{:,b}^T$, respectively, represent the a-th row vector and the b-th principal singular vector (the b-th column of $D^T$) of the matrix K. We then define the 2-norm projection of each node, including each protein, gene, lncRNA, and miRNA, in real GWGEN, to the top 85% (H) singular vectors in the following:

$$T(a) = \sqrt{\sum_{i=1}^{H} R^2(a, b)}$$
$$\text{for a} = 1, \ldots, S^* + T^* + U^* + V^* \tag{34}$$

After the projection in (34), we extracted the core GWGENs of periodontitis and healthy control; that is, the network consisted of the top 6000 ranked nodes. We uploaded the nodes of core GWGEN, in the form of an official gene id list, onto the DAVID core significant pathways by annotation of KEGG pathways. We then compared the difference between periodontitis core signaling pathways and healthy control core signaling pathways in Figure 3 and their downstream cellular dysfunctions to investigate the pathological mechanism of periodontitis. Based on the pathological mechanism of periodontitis, we selected critical biomarkers as drug targets. (The code of PNP for real GWGEN is given in [97]).

*4.6. Systematic Drug Discovery for Periodontitis Treatment via DNN-Based DTI Model Prediction and Drug Design Specifications*

In this section, the first step is to employ drug target interaction databases to train the DNN-based DTI model. The databases include BIDD [98], ChEMBL [99], STITCH [100], KEGG [101], PubChem [102], and UniProt [103]. Drug features include molecular docking, drug or gene constitutional description, amino acid composition, and more. We converted the feature sequences of drugs and targets into feature vectors via the PROFEAT website and the PyBioMed package in the Python 3.7 environment. The converted drug and target descriptor are combined below:

$$n_{drug-target} = [D, T] = [d_1, d_2, \ldots, d_N, , t_1, t_2, \ldots t_P] \tag{35}$$

where $n_{drug-target}$ is the drug-target pair in the form of a feature vector. D stands for the feature vector of the corresponding drug, and T denotes the feature vector of the target. N is the total drug feature number, and P is the total drug target feature number.

Since a large portion of raw training data has unknown interactions or negative data, the next step is data preprocessing. In order to avoid the imbalance problem, we reduced the number of unknown interaction data points. Then, for the variation of units between different features, we standardized each feature vector and normalized the significance between them. The mathematical formulas for normalization of drug and target features are respectively shown below:

$$d_i^* = \frac{d_i - \mu_i}{\sigma_i}, \ \forall i = 1, \ldots X \tag{36}$$

$$t_i^* = \frac{t_j - \mu_j}{\sigma_j}, \ \forall j = 1, \ldots Y \tag{37}$$

$d_i$ and $t_j$, respectively, represent the i-th drug and j-th target feature before standardization, where $d_i^*$ and $t_i^*$ are post-standardization results. $X$ and $Y$ denote the total numbers of drug and target features, respectively. The PCA method is implemented to avoid potential complex computing situations in the proposed DNN-based DTI model training using DTI data.

After data preprocessing, we utilized the fundamental training concept of DNN to train the DNN-based DTI model (Figure 3). In the architecture of the DNN-based DTI model, each layer can be simplified into the following functions:

$$h_n = \sigma(wx_n + b) \tag{38}$$

where $\sigma$ is the activation function by which ReLU is used for the hidden layer and the sigmoid function is used for the output layer. $x_n$ denotes the input, and $h_n$ denotes the output of the n-th drug target feature vector. $w$ stands for the weighting vector, and b is the bias vector. We selected binary-cross entropy as the cost function to calculate model loss to deal with the binary-classification problem:

$$C_n(p_n, \hat{p}_n) = -[p_n \log \hat{p}_n + (1 - p_n) \log(1 - \hat{p}_n)] \tag{39}$$

$$L(w,b) = \frac{1}{N}\sum_{n=1}^{N}C_n(p_n, \hat{p}_n) \tag{40}$$

where $p_n$ represents the true probability of positive interaction in the n-th sample and $\hat{p}_n$ represents the prediction probability of positive interaction in the n-th sample. $1 - p_n$ indicates the true probability of negative interaction, and $1 - \hat{p}_n$ indicates the prediction probability of negative interaction in the n-th sample. $L(w,b)$ denotes the average of the total loss $C_n(p_n, \hat{p}_n)$, where N is the total number of training data. Based on the cost function in (39), we applied the backward propagation algorithm to update the parameter vector θ of the weight vector $w$ and the bias vector $b$ in (41). The gradient iterative algorithm is shown as follows:

$$\theta = \begin{bmatrix} w \\ b \end{bmatrix} \tag{41}$$

$$\theta^* = arg\min_{\theta}L(\theta) \tag{42}$$

$$\theta^k = \theta^{k-1} - \eta\nabla L\left(\theta^{k-1}\right) \tag{43}$$

$$\nabla L\left(\theta^{k-1}\right) = \begin{bmatrix} \frac{\partial L\left(\theta^{k-1}\right)}{\partial w} \\ \frac{\partial L\left(\theta^{k-1}\right)}{\partial b} \end{bmatrix} \tag{44}$$

where $k$ is the k-th iteration of the DNN learning process after the training of the k-th drug-target feature vector. $\eta$ represents the learning rate, and $\nabla L\left(\theta^{k-1}\right)$ is the gradient of $L\left(\theta^{k-1}\right)$. The backward propagation method can help with the adaptation of the DNN-based DTI model parameters to fit the drug-target interaction data for each iteration. We set the learning rate $\eta$ to 0.001 and set both epochs and batch size to 100. (The code for DNN-based DTI model training is given in [97]).

Furthermore, we performed the 5-fold cross-validation strategy to verify the stability and performance of the DNN-based DTI model. We divided the training data into five equal folds, then took one as the validation data at a time. The remaining four folds were used as training data until each fold was used as validation data. To assess our network's tolerance to input change, we utilized dropout and regularization techniques by adding a dropout layer to help prevent overfitting and improve the DNN's ability to generalize new input data. We obtained data from the genome-wide microarray GSE10334 dataset, which is of high quality and provides sufficient information to solve the problem. After finishing the training of the DNN-based DTI model, we adopted the receiver operating characteristic (ROC) curve and the area under the curve (AUC) to measure and verify model performance (Figure 7). The equations for ROC curves and AUC are shown below:

$$\text{TPR (True Positive Rate)} = \frac{TP}{TP + FN} \tag{45}$$

$$\text{specificity} = \frac{TN}{TN + FP} \tag{46}$$

$$\text{FPR(Rate)} = 1 - \text{specificity} = \frac{FP}{TN + FP} \tag{47}$$

where true positive (TP) indicates that the real value is true and is identified correctly. True negative (TN) indicates that the real value is true but was identified mistakenly. False positive (FP) indicates that the real value is false and is identified correctly. False negative (FN) indicates that the real value is false but was identified mistakenly.

## 5. Conclusions

We investigated the pathological mechanism of periodontitis through systems biology methods by constructing the genome-wide genetic and epigenetic network (GWGEN) based on the corresponding microarray data. We obtained the core signaling pathways to investigate the pathological mechanism of periodontitis by applying the PNP method and KEGG pathway annotation. Significant biomarkers were identified as drug targets based on the investigation of the abnormal functions in the core signaling pathways and the downstream cellular dysfunction. With the help of a prediction DNN-based DTI model, we discovered candidate drugs for significant biomarkers of periodontitis treatment. Based on regulation ability, low drug sensitivity, and low toxicity as drug design specifications, we filtered out the potential drugs brucine, disulfiram, verapamil, and PK-11195 from candidate drugs with significant biomarkers. We combined them as a multiple-molecule drug for the therapeutic treatment of periodontitis.

Brucine demonstrated anti-inflammatory properties in animal studies [84], which have the potential to act as an inhibitor of the inflammatory effect of periodontitis. Disulfiram has been found to have antibacterial properties against various strains of bacteria [104], and further studies may prove its relationship with dental plaque bacteria. Verapamil is commonly used to treat high blood pressure [105]. While periodontitis is considered a potential risk factor for high blood pressure [106], there may also be a link between Verapamil and periodontitis. PK-11195 is strongly linked to inflammation and apoptosis [107], which are important factors in periodontitis mechanisms. However, the central nervous system toxicity of brucine is a potential limitation to its clinical application [83]. Acute disulfiram overdose in adults and children may cause hypotension, tachycardia, and dyspnea [108]. Taking too much verapamil results in disturbances of cardiac conduction, depressed myocardial contractility, and hypotension [109]. Therefore, the toxicity and side effects of the candidate drugs should be considered carefully. Recently, the combination of metronidazole and doxycycline has been found to be effective in reducing the depth of periodontal pockets and improving the overall health of the gums [110]. Metronidazole is an antibiotic that targets anaerobic bacteria, which are often found in periodontal pockets [111]. It is effective against a wide range of bacteria, including some that are resistant to other antibiotics. Doxycycline is a broad-spectrum antibiotic that targets both aerobic and anaerobic bacteria [112]. It has anti-inflammatory properties that make it effective in reducing the inflammation associated with periodontitis [113]. However, antibiotic resistance is a growing concern, and overuse of antibiotics can contribute to the development of resistant bacteria [114]. Therefore, it is important to use antibiotics judiciously and only when necessary to maximize their effectiveness and minimize their potential side effects. In conclusion, we utilized the systems biology method to describe and understand the operation of complex biological systems. Next, we develop a predictive DNN-based DTI model to predict the candidate molecular drugs for significant biomarkers in the therapeutic treatment of periodontitis. Finally, we filter out potential molecular drugs by drug design specifications, from which a multiple-molecule drug can be selected for potential therapeutic treatment for periodontitis. Systems biology combined with machine learning methods gives us an efficient approach to systematic discovery, drug discovery, and treatment design for periodontitis. We hope that this systematic medical study can provide potentially beneficial materials for future drug discovery technologies.

**Author Contributions:** Conceptualization, C.-T.W. and B.-S.C.; data curation, C.-T.W.; formal analysis, C.-T.W. and B.-S.C.; funding acquisition, B.-S.C.; investigation, C.-T.W. and B.-S.C.; software, C.-T.W. and B.-S.C.; supervision, B.-S.C.; validation, C.-T.W. and B.-S.C.; visualization, C.-T.W. and B.-S.C.; writing—original draft, C.-T.W.; writing—review and editing, C.-T.W. and B.-S.C. All authors have read and agreed to the published version of the manuscript.

**Funding:** This research received no external funding.

**Institutional Review Board Statement:** Not applicable.

**Informed Consent Statement:** Not applicable.

**Data Availability Statement:** The RNA-seq datasets of periodontitis patients are accessed from GSE10334 (https://www.ncbi.nlm.nih.gov/geo/query/acc.cgi?acc=GSE10334, accessed on 17 November 2022). The drug regulation ability data are from Phase I L1000 Level 5 datasets (https://www.ncbi.nlm.nih.gov/geo/query/acc.cgi?acc=GSE92742, accessed on 1 November 2021). The drug sensitivity datasets are from the DepMapPRISM primary screen datasets (https://depmap.org/repurposing/, accessed on 1 November 2021).

**Conflicts of Interest:** The authors declare no conflict of interest.

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
