# Peer review of "Drug Discovery for Periodontitis Treatment Based on Big Data Mining, Systems Biology, and Deep Learning Methods"

_2674-0583, doi:10.3390/synbio1010009_

Round 1

Reviewer 1 Report

Rviewer.

Manuscript ID: synbio-2348690
Type of manuscript: Article
Title: Drug Discovery for Periodontitis Based on Big Data Mining, Systems
Biology and Deep Learning Methods
Authors: Chun-Tse Wang, Bor-Sen Chen *

The aim of this review is to study the pathogenic mechanism of periodontitis using recent methods of biological analysis. The objective is to consider a therapeutic option for periodontitis via the prediction of the DTI ( drug-target interaction )  model based on the DNN

( deep neural network)   and drug design specifications. The goal is to develop an effective drug design for periodontitis before clinical trials.

The model to be considered remains theoretical, essentially based on statistical calculations. Computer algorithms  predict the interactions be tween drugs and targets (biomarkers).    It requires reservations regarding the in vivo conditions of periodontal disease.

Most basic molecular and meta-omics reports on the oral microbiome regarding health and disease have traditionally relied on small cohorts. This is because sequencing and other high-throughput techniques are expensive. Effective risk assessment for individual and specific vulnerable populations may require longitudinal approaches. This approach is a key to identify valid biomarkers of periodontitis in real time (pre and post disease). Thus, the use of larger cohorts in combination with longitudinal sampling may reveal the shared individual specific weight of environment and genetics on the oral microbiome. In the future, global surveys of diverse human populations on a larger scale may provide important clues to better understand the influence of cultural and genetic factors affecting health and periodontitis. It seems difficult under these conditions to establish a standardization of the treatment of periodontal disease.

 The article does not cover all possible aspects of the treatment of periodontal disease. Haque et al. Advances in novel therapeutic approaches for periodontal diseases. BMC Oral Health (2022) 22:492 https://doi.org/10.1186/s12903-022-02530-6

However, it requires the addition of a more recent bibliography at several levels.

 Tsuchida, S.; Nakayama, T. Metabolomics Research in Periodontal Disease byMass Spectrometry. Molecules 2022, 27, 2864. https:// doi.org/10.3390/molecules27092864

Long H, Yan L, Pu J, Liu Y, Zhong X, Wang H, Yang L, Lou F, Luo S, Zhang Y, Liu Y, Xie P, Ji P and Jin X (2022) Multi-omics analysis reveals the effects of microbiota on oral homeostasis. Front. Immunol. 13:1005992. doi: 10.3389/fimmu.2022.1005992

Text-level details .

Line 8:  Periodontitis is an infectious disease related to bacteria but also to archaea, viruses and eukaryotic organisms (cultivated and uncultivated).

Line 47: add ref (osteoporose, arteriosclerose…)

Line 49: more recent ref is necessary.

Line 52: more recent ref is necessary.

Line 61: Refer to an updated classification of periodontal diseases.

Papapanou PN, Sanz M, et al. Periodontitis: Consensus report of workgroup 2 of the 2017 World Workshop on the Classification of Periodontal and Peri-Implant Diseases

and Conditions. J Periodontol. 2018;89(Suppl 1):S173–S182.

Line 75: More recent ref is necessary.

Line 88: More recent ref is necessary.

Line 112: abbreviations must be translated in full at the beginning of the article. Otherwise a glossary is desirable at the end of the article. (fructooligosaccharides, …)

Line 119: this entire paragraph requires several references in order to validate this approach.

Line 189: and many other diseases…

Line 200: Fusobacterium nucleatum and Porphyromonas gingivalis 

Line 238: you quote before IL3 and EGF in abbreviated form and here in full? 

Line 264: FOS and L. plantarum ST-III effectively elevated the proportion of beneficial bacteria in the

intestine. Cui, S.; Guo,W.; Chen, C.; Tang, X.; Zhao, J.; Mao, B.; Zhang, H.Metagenomic Analysis of the Effectsof Lactiplantibacillus plantarum andFructooligosaccharides (FOS) on theFecal Microbiota Structure in Mice.Foods 2022, 11, 1187. https://doi.org/10.3390/foods11091187

Line 265:  More recent ref.Kizilirmak C, Bianchi ME and Zambrano S (2022) Insights on

the NF-kB System Using Live Cell Imaging: Recent Developments and Future Perspectives.

Front. Immunol. 13:886127. doi: 10.3389/fimmu.2022.886127

Line 267 : A Review of FoxO1-Metabolic Diseases and Related Drug Discoveries

Shiming Peng 1, Wei Li 2, Nannan Hou 1 and Niu Huang 1,3,*

Line 401:   Lu L, Huang R, Wu Y, Jin J-M,

Chen H-Z, Zhang L-J and Luan X (2020)  Brucine: A Review of Phytochemistry, Pharmacology, and Toxicology. Front. Pharmacol. 11:377. doi: 10.3389/fphar.2020.00377

Rviewer.

Manuscript ID: synbio-2348690
Type of manuscript: Article
Title: Drug Discovery for Periodontitis Based on Big Data Mining, Systems
Biology and Deep Learning Methods
Authors: Chun-Tse Wang, Bor-Sen Chen *

The aim of this review is to study the pathogenic mechanism of periodontitis using recent methods of biological analysis. The objective is to consider a therapeutic option for periodontitis via the prediction of the DTI ( drug-target interaction )  model based on the DNN

( deep neural network)   and drug design specifications. The goal is to develop an effective drug design for periodontitis before clinical trials.

The model to be considered remains theoretical, essentially based on statistical calculations. Computer algorithms  predict the interactions be tween drugs and targets (biomarkers).    It requires reservations regarding the in vivo conditions of periodontal disease.

Most basic molecular and meta-omics reports on the oral microbiome regarding health and disease have traditionally relied on small cohorts. This is because sequencing and other high-throughput techniques are expensive. Effective risk assessment for individual and specific vulnerable populations may require longitudinal approaches. This approach is a key to identify valid biomarkers of periodontitis in real time (pre and post disease). Thus, the use of larger cohorts in combination with longitudinal sampling may reveal the shared individual specific weight of environment and genetics on the oral microbiome. In the future, global surveys of diverse human populations on a larger scale may provide important clues to better understand the influence of cultural and genetic factors affecting health and periodontitis. It seems difficult under these conditions to establish a standardization of the treatment of periodontal disease.

 The article does not cover all possible aspects of the treatment of periodontal disease. Haque et al. Advances in novel therapeutic approaches for periodontal diseases. BMC Oral Health (2022) 22:492 https://doi.org/10.1186/s12903-022-02530-6

However, it requires the addition of a more recent bibliography at several levels.

 Tsuchida, S.; Nakayama, T. Metabolomics Research in Periodontal Disease byMass Spectrometry. Molecules 2022, 27, 2864. https:// doi.org/10.3390/molecules27092864

Long H, Yan L, Pu J, Liu Y, Zhong X, Wang H, Yang L, Lou F, Luo S, Zhang Y, Liu Y, Xie P, Ji P and Jin X (2022) Multi-omics analysis reveals the effects of microbiota on oral homeostasis. Front. Immunol. 13:1005992. doi: 10.3389/fimmu.2022.1005992

Text-level details .

Line 8:  Periodontitis is an infectious disease related to bacteria but also to archaea, viruses and eukaryotic organisms (cultivated and uncultivated).

Line 47: add ref (osteoporose, arteriosclerose…)

Line 49: more recent ref is necessary.

Line 52: more recent ref is necessary.

Line 61: Refer to an updated classification of periodontal diseases.

Papapanou PN, Sanz M, et al. Periodontitis: Consensus report of workgroup 2 of the 2017 World Workshop on the Classification of Periodontal and Peri-Implant Diseases

and Conditions. J Periodontol. 2018;89(Suppl 1):S173–S182.

Line 75: More recent ref is necessary.

Line 88: More recent ref is necessary.

Line 112: abbreviations must be translated in full at the beginning of the article. Otherwise a glossary is desirable at the end of the article. (fructooligosaccharides, …)

Line 119: this entire paragraph requires several references in order to validate this approach.

Line 189: and many other diseases…

Line 200: Fusobacterium nucleatum and Porphyromonas gingivalis 

Line 238: you quote before IL3 and EGF in abbreviated form and here in full? 

Line 264: FOS and L. plantarum ST-III effectively elevated the proportion of beneficial bacteria in the

intestine. Cui, S.; Guo,W.; Chen, C.; Tang, X.; Zhao, J.; Mao, B.; Zhang, H.Metagenomic Analysis of the Effectsof Lactiplantibacillus plantarum andFructooligosaccharides (FOS) on theFecal Microbiota Structure in Mice.Foods 2022, 11, 1187. https://doi.org/10.3390/foods11091187

Line 265:  More recent ref.Kizilirmak C, Bianchi ME and Zambrano S (2022) Insights on

the NF-kB System Using Live Cell Imaging: Recent Developments and Future Perspectives.

Front. Immunol. 13:886127. doi: 10.3389/fimmu.2022.886127

Line 267 : A Review of FoxO1-Metabolic Diseases and Related Drug Discoveries

Shiming Peng 1, Wei Li 2, Nannan Hou 1 and Niu Huang 1,3,*

Line 401:   Lu L, Huang R, Wu Y, Jin J-M,

Chen H-Z, Zhang L-J and Luan X (2020)  Brucine: A Review of Phytochemistry, Pharmacology, and Toxicology. Front. Pharmacol. 11:377. doi: 10.3389/fphar.2020.00377

Rviewer.

Manuscript ID: synbio-2348690
Type of manuscript: Article
Title: Drug Discovery for Periodontitis Based on Big Data Mining, Systems
Biology and Deep Learning Methods
Authors: Chun-Tse Wang, Bor-Sen Chen *

The aim of this review is to study the pathogenic mechanism of periodontitis using recent methods of biological analysis. The objective is to consider a therapeutic option for periodontitis via the prediction of the DTI ( drug-target interaction )  model based on the DNN

( deep neural network)   and drug design specifications. The goal is to develop an effective drug design for periodontitis before clinical trials.

The model to be considered remains theoretical, essentially based on statistical calculations. Computer algorithms  predict the interactions be tween drugs and targets (biomarkers).    It requires reservations regarding the in vivo conditions of periodontal disease.

Most basic molecular and meta-omics reports on the oral microbiome regarding health and disease have traditionally relied on small cohorts. This is because sequencing and other high-throughput techniques are expensive. Effective risk assessment for individual and specific vulnerable populations may require longitudinal approaches. This approach is a key to identify valid biomarkers of periodontitis in real time (pre and post disease). Thus, the use of larger cohorts in combination with longitudinal sampling may reveal the shared individual specific weight of environment and genetics on the oral microbiome. In the future, global surveys of diverse human populations on a larger scale may provide important clues to better understand the influence of cultural and genetic factors affecting health and periodontitis. It seems difficult under these conditions to establish a standardization of the treatment of periodontal disease.

 The article does not cover all possible aspects of the treatment of periodontal disease. Haque et al. Advances in novel therapeutic approaches for periodontal diseases. BMC Oral Health (2022) 22:492 https://doi.org/10.1186/s12903-022-02530-6

However, it requires the addition of a more recent bibliography at several levels.

 Tsuchida, S.; Nakayama, T. Metabolomics Research in Periodontal Disease byMass Spectrometry. Molecules 2022, 27, 2864. https:// doi.org/10.3390/molecules27092864

Long H, Yan L, Pu J, Liu Y, Zhong X, Wang H, Yang L, Lou F, Luo S, Zhang Y, Liu Y, Xie P, Ji P and Jin X (2022) Multi-omics analysis reveals the effects of microbiota on oral homeostasis. Front. Immunol. 13:1005992. doi: 10.3389/fimmu.2022.1005992

Text-level details .

Line 8:  Periodontitis is an infectious disease related to bacteria but also to archaea, viruses and eukaryotic organisms (cultivated and uncultivated).

Line 47: add ref (osteoporose, arteriosclerose…)

Line 49: more recent ref is necessary.

Line 52: more recent ref is necessary.

Line 61: Refer to an updated classification of periodontal diseases.

Papapanou PN, Sanz M, et al. Periodontitis: Consensus report of workgroup 2 of the 2017 World Workshop on the Classification of Periodontal and Peri-Implant Diseases

and Conditions. J Periodontol. 2018;89(Suppl 1):S173–S182.

Line 75: More recent ref is necessary.

Line 88: More recent ref is necessary.

Line 112: abbreviations must be translated in full at the beginning of the article. Otherwise a glossary is desirable at the end of the article. (fructooligosaccharides, …)

Line 119: this entire paragraph requires several references in order to validate this approach.

Line 189: and many other diseases…

Line 200: Fusobacterium nucleatum and Porphyromonas gingivalis 

Line 238: you quote before IL3 and EGF in abbreviated form and here in full? 

Line 264: FOS and L. plantarum ST-III effectively elevated the proportion of beneficial bacteria in the

intestine. Cui, S.; Guo,W.; Chen, C.; Tang, X.; Zhao, J.; Mao, B.; Zhang, H.Metagenomic Analysis of the Effectsof Lactiplantibacillus plantarum andFructooligosaccharides (FOS) on theFecal Microbiota Structure in Mice.Foods 2022, 11, 1187. https://doi.org/10.3390/foods11091187

Line 265:  More recent ref.Kizilirmak C, Bianchi ME and Zambrano S (2022) Insights on

the NF-kB System Using Live Cell Imaging: Recent Developments and Future Perspectives.

Front. Immunol. 13:886127. doi: 10.3389/fimmu.2022.886127

Line 267 : A Review of FoxO1-Metabolic Diseases and Related Drug Discoveries

Shiming Peng 1, Wei Li 2, Nannan Hou 1 and Niu Huang 1,3,*

Line 401:   Lu L, Huang R, Wu Y, Jin J-M,

Chen H-Z, Zhang L-J and Luan X (2020)  Brucine: A Review of Phytochemistry, Pharmacology, and Toxicology. Front. Pharmacol. 11:377. doi: 10.3389/fphar.2020.00377

Author Response

This is our point by point responses to the reviewers’ comments, a revised manuscript with red color to highlight the changesFurther, the revised manuscript has been edited by Grammarly, hope this revised version could meet the requirement of Synthetic Biology.

Reviewer 1 note#1: The model to be considered remains theoretical, essentially based on statistical calculations. Computer algorithms predict the interactions between drugs and targets (biomarkers).    It requires reservations regarding the in vivo conditions of periodontal disease.

Most basic molecular and meta-omics reports on the oral microbiome regarding health and disease have traditionally relied on small cohorts. This is because sequencing and other high-throughput techniques are expensive. Effective risk assessment for individual and specific vulnerable populations may require longitudinal approaches. This approach is a key to identify valid biomarkers of periodontitis in real time (pre and post disease). Thus, the use of larger cohorts in combination with longitudinal sampling may reveal the shared individual specific weight of environment and genetics on the oral microbiome. In the future, global surveys of diverse human populations on a larger scale may provide important clues to better understand the influence of cultural and genetic factors affecting health and periodontitis. It seems difficult under these conditions to establish a standardization of the treatment of periodontal disease.

Author response#1: Thank you for reviewing this manuscript. It is certain that the use of larger longitudinal samples is highly required in order to obtain more integrated results. Global surveys on more diverse human genetics are also required in the future. Although it seems difficult to establish a standardization of the treatment of periodontal disease currently, we believe that our research provides beneficial materials for future drug design technology of the disease.

Reviewer 1 note#2: The article does not cover all possible aspects of the treatment of periodontal disease.

Author response#2: Thank you for your suggestion. We added the following lines in the Introduction section: On the other hand, with the increase in antibiotic resistance among periodontal pathogens, the primary goal of periodontitis therapy is shifted to restoring homeostasis in oral microbiota and its harmonious balance with the host periodontal tissue.

The following research is cited in the revised manuscript:

 Haque et al. Advances in novel therapeutic approaches for periodontal diseases. BMC Oral Health (2022) 22:492 https://doi.org/10.1186/s12903-022-02530-6

Reviewer 1 note#3:

Text-level details.

Line 8:  Periodontitis is an infectious disease related to bacteria but also to archaea, viruses and eukaryotic organisms (cultivated and uncultivated).

Line 47: add ref (osteoporose, arteriosclerose…)

Line 49: more recent ref is necessary.

Line 52: more recent ref is necessary.

Line 61: Refer to an updated classification of periodontal diseases.

Papapanou PN, Sanz M, et al. Periodontitis: Consensus report of workgroup 2 of the 2017 World Workshop on the Classification of Periodontal and Peri-Implant Diseases

and Conditions. J Periodontol. 2018;89(Suppl 1):S173–S182.

Line 75: More recent ref is necessary.

Line 88: More recent ref is necessary.

Line 112: abbreviations must be translated in full at the beginning of the article. Otherwise a glossary is desirable at the end of the article. (fructooligosaccharides, …)

Line 119: this entire paragraph requires several references in order to validate this approach.

Line 189: and many other diseases…

Line 200: Fusobacterium nucleatum and Porphyromonas gingivalis 

Line 238you quote before IL3 and EGF in abbreviated form and here in full? 

Line 264: FOS and L. plantarum ST-III effectively elevated the proportion of beneficial bacteria in the

intestine. Cui, S.; Guo,W.; Chen, C.; Tang, X.; Zhao, J.; Mao, B.; Zhang, H.Metagenomic Analysis of the Effectsof Lactiplantibacillus plantarum andFructooligosaccharides (FOS) on theFecal Microbiota Structure in Mice.Foods 2022, 11, 1187. https://doi.org/10.3390/foods11091187

Line 265:  More recent ref.Kizilirmak C, Bianchi ME and Zambrano S (2022) Insights on

the NF-kB System Using Live Cell Imaging: Recent Developments and Future Perspectives.

Front. Immunol. 13:886127. doi: 10.3389/fimmu.2022.886127

Line 267 : A Review of FoxO1-Metabolic Diseases and Related Drug Discoveries

Shiming Peng 1, Wei Li 2, Nannan Hou 1 and Niu Huang 1,3,*

Line 401:   Lu L, Huang R, Wu Y, Jin J-M,

Chen H-Z, Zhang L-J and Luan X (2020)  Brucine: A Review of Phytochemistry, Pharmacology, and Toxicology. Front. Pharmacol. 11:377. doi: 10.3389/fphar.2020.00377

Author response#3: Thank you very much for your suggestions on modifications of the manuscript. We thoroughly went through the manuscript and revised it based on your review comments. The references that you suggested were cited in the revised version.

Reviewer 2 Report

The authors tackle an important but often understudied health issue - periodontitis - using network data mining approaches to connect relationships from public databases.

The objective is laudable and the overall study design is appropriate. However, there are some technical and domain details that need to be added to improve reproducibility and utility of the product.

Firstly, the pathways identified by the authors in Figure 1 are not entirely novel. These align well with what domain researchers expect in the field.  A greater contribution from the method would be to provide relative ranking information of the important nodes in the pruned network.  Perhaps and unsupervised rank aggregation metric similar to those used in knowledge graphs would be helpful.  Or if the authors could provide other metrics in their own algorithm utilized to determine the relevance of these nodes, it would better support their results.  Namely, what was the relevance of the selected nodes versus the average relevance of nodes in the larger, unpruned network?

Secondly, more information on the subjects' data is necessary to compare what is considered a "healthy" control versus a "diseased sample".  Please provide basic descriptive statistics of each overall populations (age, gender, etc.) and any metrics regarding periodontal disease duration, severity, and method of diagnosis.

Third, more information is necessary on the pruning process and the parameter optimization for detecting false positives. In particular, there were no metrics provided to determine how stable the results are - whether a sensitivity analysis, assessment that includes injection of noise, oversampling with synthetic patients or anything to better assess the network stability to the identified solutions.  The authors state 5-fold cross-validation was used to assess stability, but this is insufficient and inherently biased.  It is difficult to determine if this is a truly generalizable solution given the relatively small number of patient samples (about 20 patients per fold) without assessing how well the network tolerates input changes.

Next, the authors should state  differentiates this method compared to other drug repurposing methods, like molecular targeting or knowledge graph network approaches that can identify related drugs and conditions. For example, a recent cross-domain knowledge graph text mining algorithm determined tyrosine kinase inhibitors that treat BCR-ABL chronic myeloid leukemia are strongly associated with periodontitis, which was a finding not previously widely accepted: https://www.mdpi.com/2072-6694/14/19/4686. Notably, this pathway is implicitly represented in the authors' present work as a contributor to periodontitis.

Finally, the authors do not provide domain context for WHY their chosen drugs could be good candidates for periodontitis.  The authors should expend more discussion on potential mechanisms for how the selected drugs map back to specific mechanisms in their network in addition to similar clinical findings.  The present Discussion for the selected candidates is very superficial and too general.

OTHER: The manuscript suffers from moderate to extensive English language writing faults that impact the manuscript clarity and publication quality.  While some typos and grammar issues are expected in any manuscript draft, this draft far exceeds the number of written English deficiencies that would be expected for a journal article in MDPI.

Significant English editing required

Author Response

This is our point by point responses to the reviewers’ comments, a revised manuscript with red color to highlight the changesFurther, the revised manuscript has been edited by Grammarly, hope this revised version could meet the requirement of Synthetic Biology.

Reviewer 2 note#1: Firstly, the pathways identified by the authors in Figure 1 are not entirely novel. These align well with what domain researchers expect in the field. A greater contribution from the method would be to provide relative ranking information of the important nodes in the pruned network.  Perhaps and unsupervised rank aggregation metric similar to those used in knowledge graphs would be helpful. Or if the authors could provide other metrics in their own algorithm utilized to determine the relevance of these nodes, it would better support their results. Namely, what was the relevance of the selected nodes versus the average relevance of nodes in the larger, unpruned network?

Author response#1: Thank you for your comment and suggestions. We updated a new table: Table 1 (B), which listed the top four pathways based on the enriched DAVID analysis of periodontitis and healthy control with low P-value. The listed KEGG pathways in Table 1 (B) were utilized for the construction of the common and specific core signaling pathways of periodontitis and non-periodontitis as shown in Figure 3.

Reviewer 2 note#2: Secondly, more information on the subjects' data is necessary to compare what is considered a "healthy" control versus a "diseased sample".  Please provide basic descriptive statistics of each overall populations (age, gender, etc.) and any metrics regarding periodontal disease duration, severity, and method of diagnosis.

Author response#2: Thank you for your suggestions. We added the following lines in section 4.2: We downloaded genome-wide microarray data from the National Center for Biotechnology Information (NCBI). The genome-wide microarray data include diseased and healthy gingival tissues of 90 non-smoking patients (63 with chronic and 27 with aggressive periodontitis). The patients had no history of systematic periodontal therapy other than occasional prophylaxis, and had received no systemic antibiotics or anti-inflammatory drugs for more than 6 months. Furthermore, the patients did not have diabetes or any systemic condition that entails a diagnosis of “Periodontitis as a manifestation of systemic diseases”.

Reviewer 2 note#3: Third, more information is necessary on the pruning process and the parameter optimization for detecting false positives. In particular, there were no metrics provided to determine how stable the results are - whether a sensitivity analysis, assessment that includes injection of noise, oversampling with synthetic patients or anything to better assess the network stability to the identified solutions. The authors state 5-fold cross-validation was used to assess stability, but this is insufficient and inherently biased.  It is difficult to determine if this is a truly generalizable solution given the relatively small number of patient samples (about 20 patients per fold) without assessing how well the network tolerates input changes.

Author response#3: Thank you for your comments. The pruning process and parameter optimization for detecting false positives are described as follows: In general, increasing parameter number (system order) will result in good model fit such that log residual error in the first term of AIC decreases and the second term of AIC increases, and vice versa. Therefore, there should be exact parameter numbers of system order to achieve the minimum AIC among all possible combinations for each protein and gene. Forward and backward stepwise algorithms were both adopted to gain the minimum AIC in equations (21) ~ (26) with the help of lsqlin function in 2021 MATLAB optimization toolbox. Therefore, we trimmed the insignificant parameters as false positives in the candidate GWGEN out of system order detected by AIC to obtain real GWGENs of periodontitis and healthy control. (added in lines 645 ~ 653)

To assess our network's tolerance to input change, we utilized dropout and regularization techniques by adding dropout layer to help prevent overfitting and improve the DNN’s ability to generalize a new input data. We obtained data from the genome-wide microarray GSE10334 dataset, which is of high quality and sufficient information to the problem. (added in lines 783 ~ 787)

Reviewer 2 note#4: Next, the authors should state differentiates this method compared to other drug repurposing methods, like molecular targeting or knowledge graph network approaches that can identify related drugs and conditions. For example, a recent cross-domain knowledge graph text mining algorithm determined tyrosine kinase inhibitors that treat BCR-ABL chronic myeloid leukemia are strongly associated with periodontitis, which was a finding not previously widely accepted: https://www.mdpi.com/2072-6694/14/19/4686. Notably, this pathway is implicitly represented in the authors' present work as a contributor to periodontitis.

Author response#4: We added the following lines in the Discussion section:

Our method strongly differentiates with other drug repurposing methods. For example, the paper titled "Cross-Domain Text Mining to Predict Adverse Events from Tyrosine Kinase Inhibitors for Chronic Myeloid Leukemia" in [82] explores the use of text mining to predict adverse events associated with Tyrosine Kinase Inhibitors (TKIs) used to treat Chronic Myeloid Leukemia (CML). The authors propose a cross-domain approach, utilizing data from multiple sources including electronic health records, social media, and scientific literature to improve the accuracy of adverse event prediction. Our method focuses on building the pathogenic network of periodontitis based on big data mining of PPIN and GRN database and microarray data of periodontitis and healthy controls via reverse engineering techniques by Systems Biology approach. We compose the core signaling pathways of periodontitis and healthy control to develop a comprehensive and quantitative understanding of the pathogenic mechanism by studying their components, interactions, and emergent properties. This approach involves the use of computational models and system identification (reverse engineering method), data-driven approaches, and experimental techniques to analyze the pathogenic mechanism of periodontitis and identify the significant biomarkers as drug targets of periodontitis treatment. Furthermore, the powerful DNN-based DTI model is trained by DTI databases to predict potential molecular drugs for the significant biomarkers. Overall, systems biology aims to provide a holistic and quantitative understanding of pathogenic mechanism of periodontitis, with the ultimate goal of improving the restoration of downstream cellular dysfunctions in periodontitis.

Reviewer 2 note#5: Finally, the authors do not provide domain context for WHY their chosen drugs could be good candidates for periodontitis.  The authors should expend more discussion on potential mechanisms for how the selected drugs map back to specific mechanisms in their network in addition to similar clinical findings.  The present Discussion for the selected candidates is very superficial and too general.

Author response#5: Thank you for your suggestion. We added the following lines in the Discussion section, according to the four candidate drugs we selected.

Brucine N-oxide, a derivative of brucine, has also demonstrated analgesic and anti-inflammatory properties in animal studies. It has been shown to inhibit the activity of certain enzymes involved in the production of inflammatory mediators, as well as reduce pain sensitivity in animals[86]. We found out that brucine is able to upregulate the expression of FOS, FOXO1 and NF-κB, and at the same time downregulate TSC2. Disulfiram is an FDA-approved drug used for the treatment of alcoholism, and it has been found to have antibacterial properties against various strains of bacteria[87]. Studies have shown that disulfiram is a possible treatment for parasitic infections, anxiety disorder, cancer and latent HIV infection[88]. Disulfiram is proven to be effective in killing various kinds of microbes, where it can be received as potential antibiotic therapy[89]. FOS and FOSB were investigated and proven as risk factors for periodontitis[90]. Moreover, FOS is a promising option to treat systematic infections. Our study discovered the ability of disulfiram to restore the expression of FOS, as well as other biomarkers back to normal expression. Verapamil is commonly used to treat high blood pressure, angina (chest pain), and certain heart rhythm disorders[91]. Recent studies based on meta-analysis revealed significant positive association between periodontitis and increased systolic and diastolic blood pressure, and then concludes that periodontitis should be considered as a potential risk factor for high blood pressure[92, 93]. Verapamil strongly measures up to our standards, as it has respectively low sensitivity and high LC50 value. PK-11195 is a synthetic compound utilized in molecular imaging studies to visualize the peripheral benzodiazepine receptor (PBR) in the body. PBR is found in immune cells, glial cells in the brain, and steroidogenic cells in the adrenal glands and gonads. PK-11195 is strongly linked to inflammation and apoptosis, which are important factors in periodontitis mechanism[94]. It is found that PK-11195 has the potential to enhance the effectiveness of chemotherapy in leukemia and myeloma cells[95]. Background studies have shown that oral manifestations, such as gingival bleeding, gingival inflammation or overgrowth, and periodontitis, are the first symptoms of leukemia[96, 97].

Reviewer 3 Report

Comments:

1. Full name of FOS, TSC2, FOXO1 and NF-κB should be included in Introduction. And initial with small letter for Brucine, Disulfiram, Verapamil in Abstract and line 116, 325, 420. Moreover, the physicochemical properties and pharmacodynamic characteristics of Brucine, Disulfiram, Verapamil and PK-11195 should be stated in Introduction.

2. Line 58:  dental check-ups..  ??

3. Literature review on dual drug delivery for periodontitis treatment should be included in Introduction.

4. The literature review on drugs employed with localized drug targeting of periodontitis should be addressed in Introduction. Type of such delivery systems should be indicated such as in situ forming gel and matrix for periodontitis treatment.

5. There was no any data below legend of Table 1,2,3.?? Thus the referee do no see the data in these tables

6. The comparison of result to inyeresting drug such as doxycycline and azithromycin is necessary or indicates in Introduction.

7. For the context of manuscript please check the meaning for using wording "periodontitis" and "periodontitis treatment".

8. More discussion should be added with supporting references in 3.

9. Importantly, the description on Brucine, Disulfiram, Verapamil and PK-11195 for their bioactivities and their dose and recommended administration related to periodontitis treatment should be included in Introduction and discussion.

10. Please indicate the possibility of above four compounds for periodontitis treatment or any point of views. and comparison with commonly used drug for periodontitis treatment with evidence-based and rationale of cost of treatment and side effect.

11. The concern on above mentioned in 10 on Conclusion should be addressed and adviced.

12. The brand, company, city, country on all program and machine in 4. should be revealed. 

13. Please avoid using "We" in manuscript.

14. Please add schematic diagram presenting your obtained finding in 3. Discussion.

The language check from third party should be undertaken

Author Response

This is our point by point responses to the reviewers’ comments, a revised manuscript with red color to highlight the changesFurther, the revised manuscript has been edited by Grammarly, hope this revised version could meet the requirement of Synthetic Biology.

Reviewer 3 note#1:

  1. Full name of FOS, TSC2, FOXO1 and NF-κB should be included in Introduction. And initial with small letter for Brucine, Disulfiram, Verapamil in Abstract and line 116, 325, 420. Moreover, the physicochemical properties and pharmacodynamic characteristics of Brucine, Disulfiram, Verapamil and PK-11195 should be stated in Introduction. 2. Line 58:  dental check-ups..  ??

Author response#1:

Thank you for your suggestions, the following lines were added in Introduction:

Brucine is a natural compound found in the seeds of the Nux-vomica plant. It has both bioactive and toxic properties and is considered a weak alkaline indole alkaloid. Recent studies in pharmacology and clinical practice have shown that brucine has a variety of potential pharmacological effects, including anti-tumor, anti-inflammatory, analgesic, and effects on the cardiovascular and nervous systems[28]. Disulfiram is a medication used in the treatment of alcoholism. The pharmacological effects of disulfiram are primarily related to its ability to produce an aversive reaction to alcohol. This helps to discourage individuals from drinking by making the experience unpleasant[29]. The pharmacological effects of verapamil are primarily related to its ability to reduce blood pressure and alleviate symptoms of angina. It can also be used to treat certain cardiac arrhythmias[30]. The pharmacological effects of PK-11195 are primarily related to its ability to modulate the immune response in the brain. It has been shown to have anti-inflammatory and neuroprotective effects, and may have potential therapeutic applications in conditions like Alzheimer's disease and Parkinson's disease[31].

Reviewer 3 note#2:

  1. Literature review on dual drug delivery for periodontitis treatment should be included in Introduction.
  2. The literature review on drugs employed with localized drug targeting of periodontitis should be addressed in Introduction. Type of such delivery systems should be indicated such as in situ forming gel and matrix for periodontitis treatment.
  3. There was no any data below legend of Table 1,2,3.?? Thus the referee do no see the data in these tables
  4. The comparison of result to inyeresting drug such as doxycycline and azithromycin is necessary or indicates in Introduction.

Author response#2:

  1. We included the following lines for dual drug delivery for periodontitis:

There are several approaches to dual drug delivery for periodontitis treatment. One method involves the use of drug-loaded nanoparticles, which can be applied directly to the affected area[16]. Another approach involves the use of drug-eluting implants, which are placed directly into the periodontal pocket[17].

  1. The following lines are included in Introduction:

Additionally, localized drug targeting such as Situ forming gel and matrix systems have been studied as potential treatments for periodontitis[16].

  1. Tables 1, 2 and 3 were added.
  2. The following lines are included in Introduction:

Although antibiotics and anti-inflammatory drugs can be effective in improving clinical outcomes and slowing down the progression of the disease, (most commonly used systemic antibiotics include amoxicillin, metronidazole and doxycycline) indiscriminate use of antibiotics can lead to the development of antibiotic-resistant strains of bacteria, as well as other adverse effects[19].

Reviewer 3 note#3:

  1. For the context of manuscript please check the meaning for using wording "periodontitis" and "periodontitis treatment".

Author response#3:

Thank you for your suggestion. We have improved using wording by checking "periodontitis" and "periodontitis treatment" in the revised version.

Reviewer 3 note#4:

  1. More discussion should be added with supporting references in 3.

Author response#4:

More discussions are added in the revised version.

Reviewer 3 note#5:

  1. Importantly, the description on Brucine, Disulfiram, Verapamil and PK-11195 for their bioactivities and their dose and recommended administration related to periodontitis treatment should be included in Introduction and discussion.

Author response#5:

Thank you for the suggestions. The description on Brucine, Disulfiram, Verapamil and PK-11195 for their bioactivities and their dose and recommended administration related to periodontitis treatment were added. (Discussion section)

Brucine N-oxide, a derivative of brucine, has also demonstrated analgesic and anti-inflammatory properties in animal studies. It has been shown to inhibit the activity of certain enzymes involved in the production of inflammatory mediators, as well as reduce pain sensitivity in animals[86]. We found out that brucine is able to upregulate the expression of FOS, FOXO1 and NF-κB, and at the same time downregulate TSC2. Disulfiram is an FDA-approved drug used for the treatment of alcoholism, and it has been found to have antibacterial properties against various strains of bacteria[87]. Studies have shown that disulfiram is a possible treatment for parasitic infections, anxiety disorder, cancer and latent HIV infection[88]. Disulfiram is proven to be effective in killing various kinds of microbes, where it can be received as potential antibiotic therapy[89]. FOS and FOSB were investigated and proven as risk factors for periodontitis[90]. Moreover, FOS is a promising option to treat systematic infections. Our study discovered the ability of disulfiram to restore the expression of FOS, as well as other biomarkers back to normal expression. Verapamil is commonly used to treat high blood pressure, angina (chest pain), and certain heart rhythm disorders[91]. Recent studies based on meta-analysis revealed significant positive association between periodontitis and increased systolic and diastolic blood pressure, and then concludes that periodontitis should be considered as a potential risk factor for high blood pressure[92, 93]. Verapamil strongly measures up to our standards, as it has respectively low sensitivity and high LC50 value. PK-11195 is a synthetic compound utilized in molecular imaging studies to visualize the peripheral benzodiazepine receptor (PBR) in the body. PBR is found in immune cells, glial cells in the brain, and steroidogenic cells in the adrenal glands and gonads. PK-11195 is strongly linked to inflammation and apoptosis, which are important factors in periodontitis mechanism[94]. It is found that PK-11195 has the potential to enhance the effectiveness of chemotherapy in leukemia and myeloma cells[95]. Background studies have shown that oral manifestations, such as gingival bleeding, gingival inflammation or overgrowth, and periodontitis, are the first symptoms of leukemia[96, 97].

Reviewer 3 note#6:

  1. Please indicate the possibility of above four compounds for periodontitis treatment or any point of views. and comparison with commonly used drug for periodontitis treatment with evidence-based and rationale of cost of treatment and side effect. 11. The concern on above mentioned in 10 on Conclusion should be addressed and adviced.

Author response#6:

The following lines were added in Conclusion section.

However, the central nervous system toxicity of brucine is a potential limitation to its clinical application[85]. Acute disulfiram overdose in adults and children may cause hypotension, tachycardia, and dyspnea[106]. Taking too much verapamil results in disturbances of cardiac conduction, depressed myocardial contractility and hypotension[107]. Therefore, the toxicity and side effects of the candidate drugs should be considered carefully.

Round 2

Reviewer 1 Report

After verification of the requested modifications, I see no inconvenience to its publication.

Author Response

Thank you for reviewing this manuscript. Please let me know if there is any problem.

Reviewer 2 Report

The authors have made changes to address most of the key technical concerns.  

The writing (content and style) improved.  However, there are still some extraordinarily long sentences that need edited for clarity and grammar.  A final round of English editing would be suggested.

Author Response

Revised version is uploaded, marked with "Track Changes" function. Sentences that are too long were break into shorter sentences with grammar corrections, as shown in the following:

Original sentence:

To achieve our ultimate goal of designing a combination of molecular drugs for periodontitis treatment, a deep neural network (DNN)-based drug-target interaction (DTI) model was employed to be trained with the existing drug-target interaction databases for the prediction of candidate molecular drugs of the significant biomarkers.

Edited sentences:

To achieve our ultimate goal of designing a combination of molecular drugs for periodontitis treatment, a deep neural network (DNN)-based drug-target interaction (DTI) model was employed. The model is trained with the existing drug-target interaction databases for the prediction of candidate molecular drugs of the significant biomarkers.

Original sentence:

Next, we utilized the combination of system identification and system order detection by the corresponding microarray data to delete false positive interactions and regulations from candidate GWGEN to obtain the real GWGENs of periodontitis and healthy control[25].

Edited sentences:

Next, we utilized the combination of system identification and system order detection by the corresponding microarray data to delete false positive interactions and regulations from candidate GWGEN. After deleting the false positives, we obtained the real GWGENs of periodontitis and healthy control[25].

Original sentence:

We then filtered out brucine, disulfiram, verapamil and PK-11195 as potential molecular drugs according to adequate regulation ability, low toxicity and high sensitivity as drug design specifications, then a multiple molecule-drug could be selected from these four potential molecular drugs for therapeutic treatment of periodontitis from these 4 potential molecular drugs.

Edited sentences:

We then filtered out brucine, disulfiram, verapamil and PK-11195 as potential molecular drugs according to adequate regulation ability, low toxicity and high sensitivity. A multiple molecule-drug could be selected from these four potential molecular drugs for therapeutic treatment of periodontitis.

Original sentence:

Node extraction was based on their corresponding significant projection values, we selected top 6000 ranked nodes that construct the significant networks in the real GWGEN of periodontitis and healthy control.

Edited sentences:

Node extraction was based on their corresponding significant projection values. We selected top 6000 ranked nodes that construct the significant networks in the real GWGEN of periodontitis and healthy control.

Original sentence:

This approach involves the use of computational models and system identification (reverse engineering method), data-driven approaches, and experimental techniques to analyze the pathogenic mechanism of periodontitis and identify the significant biomarkers as drug targets of periodontitis treatment.

Edited sentences:

This approach involves the use of computational models and system identification (reverse engineering method), data-driven approaches, and experimental techniques to analyze the pathogenic mechanism of periodontitis.

Original sentence:

Based on regulation ability, low drug sensitivity and low toxicity as drug design specifications, we filtered out the potential drugs brucine, disulfiram, verapamil and PK-11195 from candidate drugs of significant biomarkers to combine as a multiple-molecule drug for therapeutic treatment of periodontitis.

Edited sentence:

Based on regulation ability, low drug sensitivity and low toxicity as drug design specifications, we filtered out the potential drugs brucine, disulfiram, verapamil and PK-11195 from candidate drugs of significant biomarkers. We combined them as a multiple-molecule drug for therapeutic treatment of periodontitis.

Reviewer 3 Report

Some previous comments have not been explained and undertaken.

  1. Full name of FOS, TSC2, FOXO1 and NF-κB should be included firstly before (their abbreviations) in Abstract and Introduction. 
  2. The meaning of dual drug delivery is not correct that it should be employing two active compounds as the same time such as ibuprofen with doxycycline and other two drugs for periodontitis treatment on Literature review on dual drug delivery for periodontitis treatment should be included in Introduction.
  1. The literature review on drugs employed with localized drug targeting of periodontitis should be addressed in Introduction. Type of such delivery systems should be indicated such as in situ forming gel (ISG) and matrix (ISM) for periodontitis treatment such as using bleached shellac, ethyl cellulose, lauric acid, rosin and borneol as matrix or gel forming agents. and (What is your "situ forming gel"?)[16]
  2. Please indicate the possibility of your selected four compounds for periodontitis treatment and comparison with commonly used drugs for periodontitis treatment such as metronidazole and doxycycline with evidence-based and rationale of cost of treatment and side effect. 
  1.  

 Minor editing of English language required

Author Response

Revised version is uploaded, marked with "Track Changes" function.

Reviewer 3 note 1:

Full name of FOS, TSC2, FOXO1 and NF-κB should be included firstly before (their abbreviations) in Abstract and Introduction. 

Author response 1:

Thank you for your suggestion. We modified our sentences in Abstract and Introduction as following:

Fos Proto-Oncogene, AP-1 Transcription Factor Subunit (FOS), TSC Complex Subunit 2 (TSC2), Forkhead Box O1 (FOXO1) and Nuclear factor kappa-light-chain-enhancer of activated B cells (NF-κB)

Reviewer 3 note 2:

2. The meaning of dual drug delivery is not correct that it should be employing two active compounds as the same time such as ibuprofen with doxycycline and other two drugs for periodontitis treatment on Literature review on dual drug delivery for periodontitis treatment should be included in Introduction.

3. The literature review on drugs employed with localized drug targeting of periodontitis should be addressed in Introduction. Type of such delivery systems should be indicated such as in situ forming gel (ISG) and matrix (ISM) for periodontitis treatment such as using bleached shellac, ethyl cellulose, lauric acid, rosin and borneol as matrix or gel forming agents. and (What is your "situ forming gel"?)[16]

Author response 2:

Additionally, dual drug delivery could be used for periodontal treatment. Examples include in situ forming gel (ISFG) loaded with doxycycline hyclate and ibuprofen[16], and in situ forming matrix (ISFM) loaded with vancomycin hydrochloride (VH) and borneol[17]

References:

  1. Puyathorn, N., et al., Physicochemical and Bioactivity Characteristics of Doxycycline Hyclate-Loaded Solvent Removal-Induced Ibuprofen-Based In Situ Forming Gel. Gels, 2023. 9(2): p. 128.
  2. Lertsuphotvanit, N., et al., Borneol-based antisolvent-induced in situ forming matrix for crevicular pocket delivery of vancomycin hydrochloride. International Journal of Pharmaceutics, 2022. 617: p. 121603

Reviewer 3 note 3:

4. Please indicate the possibility of your selected four compounds for periodontitis treatment and comparison with commonly used drugs for periodontitis treatment such as metronidazole and doxycycline with evidence-based and rationale of cost of treatment and side effect.

Author response 3:

We added the following lines in the Conclusion section:

Brucine demonstrated anti-inflammatory properties in animal studies, which has the potential to act as a inhibitor of the inflammatory effect of periodontitis. Disulfiram has been found to have antibacterial properties against various strains of bacteria, further studies may prove its relation with dental plaque bacteria. Verapamil is commonly used to treat high blood pressure. While periodontitis is considered a potential risk factor for high blood pressure, there may also be a link between Verapamil and periodontitis. PK-11195 is strongly linked to inflammation and apoptosis, which are important factors in periodontitis mechanism. However, the central nervous system toxicity of brucine is a potential limitation to its clinical application[85]. Acute disulfiram overdose in adults and children may cause hypotension, tachycardia, and dyspnea[106]. Taking too much verapamil results in disturbances of cardiac conduction, depressed myocardial contractility and hypotension[107]. Therefore, the toxicity and side effects of the candidate drugs should be considered carefully. Recently, the combination of metronidazole and doxycycline has been found to be effective in reducing the depth of periodontal pockets and improving the overall health of the gums[108]. Metronidazole is an antibiotic that targets anaerobic bacteria, which are often found in periodontal pockets[109]. It is effective against a wide range of bacteria, including some that are resistant to other antibiotics. Doxycycline is a broad-spectrum antibiotic that targets both aerobic and anaerobic bacteria. It has anti-inflammatory properties that make it effective in reducing the inflammation associated with periodontitis[110]. However, antibiotic resistance is a growing concern, and overuse of antibiotics can contribute to the development of resistant bacteria. Therefore, it is important to use antibiotics judiciously and only when necessary to maximize their effectiveness and minimize their potential side effects.

Round 3

Reviewer 3 Report

Please consider for possibility to include cited refs in Conclusion.

There is no heading of left side of Table 1.

Please add z-y axis, tick of Fig. 5,6,7 and move their label into figure area. 

Author Response

The revised version of the manuscript is uploaded.

Reviewer comment #1:

Please consider for possibility to include cited refs in Conclusion.

Author response #1:

Thank you for your suggestions. Cited references are were added in Conclusion.

Reviewer comment #2:

There is no heading of left side of Table 1.

Author response #2:

Thank you for your suggestions. Headings were added in Table 1(A) and (B)

Reviewer comment #3:

Please add z-y axis, tick of Fig. 5,6,7 and move their label into figure area. 

Author response #3:

Thank you for your suggestion. The axis and tick were added in Figure 5, 6 and 7.
